# Spectral and steady-state properties of fermionic random quadratic Liouvillians

João Costa[1*], Pedro Ribeiro[1,2], Andrea De Luca[3], Tomaž Prosen[4] and Lucas Sá[1]

**1** CeFEMA, Instituto Superior Técnico, Universidade de Lisboa,
Av. Rovisco Pais, 1049-001 Lisboa, Portugal
**2** Beijing Computational Science Research Center, Beijing 100193, China
**3** Laboratoire de Physique Théorique et Modélisation, CY Cergy Paris Université, CNRS,
F-95302 Cergy-Pontoise, France
**4** Department of Physics, Faculty of Mathematics and Physics, University of Ljubljana,
Jadranska 19, SI-1000 Ljubljana, Slovenia

⋆ joao.sanfins.costa@tecnico.ulisboa.pt

## Abstract

We study spectral and steady-state properties of generic Markovian dissipative systems described by quadratic fermionic Liouvillian operators of the Lindblad form. The Hamiltonian dynamics is modeled by a generic random quadratic operator, i.e., as a featureless superconductor of class D, whereas the Markovian dissipation is described by $M$ random linear jump operators. By varying the dissipation strength and the ratio of dissipative channels per fermion, $m = M/(2N_F)$, we find two distinct phases where the support of the single-particle spectrum has one or two connected components. In the strongly dissipative regime, this transition occurs for $m = 1/2$ and is concomitant with a qualitative change in both the steady-state and the spectral gap that rules the large-time dynamics. Above this threshold, the spectral gap and the steady-state purity qualitatively agree with the fully generic (i.e., non-quadratic) case studied recently. Below $m = 1/2$, the spectral gap closes in the thermodynamic limit and the steady-state decouples into an ergodic and a nonergodic sector yielding a non-monotonic steady-state purity as a function of the dissipation strength. Our results show that some of the universal features previously observed for fully random Liouvillians are generic for a sufficiently large number of jump operators. On the other hand, if the number of dissipation channels is decreased the system can exhibit nonergodic features, rendering it possible to suppress dissipation in protected subspaces even in the presence of strong system-environment coupling.

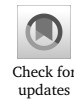

# 1 Introduction

The vast majority of systems in nature have their own time evolution deeply influenced by the interaction with their environment. Under the assumption of either a very weakly or very strongly coupled environment, with memory times much shorter than all other characteristic time scales (Markovian approximation), the time evolution equation for the system's reduced density matrix $\rho$ assumes the Gorini-Kossakowski-Sudarshan-Lindblad form [1–3], or just Lindblad form for short:

$$\frac{d\rho}{dt} = \mathcal{L}[\rho] = -i[\hat{H}, \rho] + g\sum_{\mu=1}^{M} \left(2\hat{L}_\mu \rho \hat{L}_\mu^\dagger - \{\hat{L}_\mu^\dagger \hat{L}_\mu, \rho\}\right), \tag{1}$$

where the superoperator $\mathcal{L}$ is known as the Liouvillian and $g$ is a parameter that quantifies the dissipation strength. The $M$ independent jump operators, $\hat{L}_\mu$, represent channels of interaction with the environment that act, e.g., as sources of dephasing and dissipation. In the absence of these operators, the evolution is that of a closed system, and the Liouvillian becomes just the von Neumann generator, $-i[\hat{H}, \rho]$. Note that, although for strictly zero dissipation the ensuing unitary time evolution is completely determined by the Hamiltonian, $\hat{H}$, for any finite dissipation strength, there is a (generically unique) steady state the system relaxes to at large times. Although the Markovian approximation leads to a considerable simplification of the time-evolution equation, it leaves out the possibility of studying, for instance, some quantum transport setups, for which different approaches need to be taken [4,5]. Nevertheless, it still finds many valuable applications in various subjects, namely in quantum optics and quantum computation [6].

Despite the clear simplification the Markovian approximation brings to the problem, the determination of the spectral and steady-state properties of the Liouvillian for generic Hamiltonian and jump operators remains a formidable task and is the object of intense ongoing research. A further simplification can be achieved if we restrict our analysis to quadratic systems, which are characterized by a quadratic Hamiltonian and linear jump operators in bosonic or fermionic creation and annihilation operators [7–15]. More precisely, and focusing on a system with $N_F$ complex fermions satisfying $\{c_i, c_j^\dagger\} = \delta_{ij}$, the Liouvillian of Eq. (1) is said to be quadratic if

$$\hat{H} = \frac{1}{2}\sum_{i,j=1}^{2N_F} C_i^\dagger H_{ij} C_j \qquad \text{and} \qquad \hat{L}_\mu = \sum_{j=1}^{2N_F} l_{\mu j} C_j, \tag{2}$$

with $C = \{c_1, \ldots, c_{N_F}, c_1^\dagger, \ldots, c_{N_F}^\dagger\}^T$ a vector of fermionic creation and annihilation operators that satisfies $\{C_i, C_j^\dagger\} = \delta_{ij}$. The quadratic Lindblad operator obtained from this construction ensures that the dynamics preserve the Gaussian form of an initial density matrix. Thus, the time evolution of the $2^{N_F} \times 2^{N_F}$ density matrix can be encoded by its second moments' matrix—the correlation matrix—of size $2N_F \times 2N_F$. Analogously to quadratic Hamiltonian systems, it is possible to construct a single-particle basis whose dimension scales linearly with the number of fermionic modes, $N_F$. Many-body observables, such as the Liouvillian's many-body spectrum and steady-state correlators, can be straightforwardly computed from single-particle quantities. Moreover, the single-body spectrum can be identified with that of a non-Hermitian Hamiltonian [7], leaving the determination of the Liouvillian's spectral properties only dependent on the specification of the single-particle Hamiltonian $H$ and jump operators $l$.

However, most systems of interest are extremely complex, with many degrees of freedom and exhibiting very complicated dynamics, rendering the task of determining these operators impossible in practice. We thus resort to Jayne's principle of maximal entropy [16,17], constraining these operators to a manifold compatible with their symmetries and randomizing all

other degrees of freedom. This principle has proven immensely successful for complex closed quantum systems, as pioneered by Wigner, who proposed to approximate the Hamiltonian of a compound nucleus by a random matrix [18]. The success of the approach induced a variety of different attempts to generalize it [19], culminating in the formulation of the celebrated Bohigas-Giannoni-Schmit conjecture [20] that states that the spectral correlations of quantum chaotic Hamiltonians coincide with those of a random matrix of the appropriate symmetry class. Besides level correlations, also level densities are captured by many-body random matrix models with few-body interactions (quartic in creation and annihilation operators), the random embedded ensembles [21–26] and the Sachdev-Ye-Kitaev (SYK) model [27–31]. Finally, the interplay of single-body chaos and many-body integrability in random quadratic fermionic Hamiltonians has also been studied [32–36].

More recently, the random matrix theory (RMT) approach has been extended to generic open quantum systems with Markovian dissipation [37–43]. A random Liouvillian was shown to have a lemon-shaped spectral support [37, 43] and its spectral gap was extensively studied [38]. The dependence of spectral and steady-state properties of the random Liouvillian with system size, dissipation strength, and the number of jump operators was addressed in Ref. [40]. Considering jump operators with few-body interactions has clarified the role of locality in the separation of dissipative timescales [44,45] and metastability [46] and allowed for the analytic computation of the spectral gap of the strongly-coupled SYK Liouvillian [47–50]. An important open question is to establish the universality of these results. Encouraging first steps showed that, besides local level statistics [51], the steady-state of fully random Kraus maps and Liouvillians coincide [41] and that global spectral features of quartic Liouvillians are qualitatively similar to the fully random case [47,48].

In this paper, we extend this ongoing effort and study the single-body spectral and steady-state properties of fermionic random quadratic Liouvillians. The rest of the paper is organized as follows. In Sec. 1.1 we summarize the main results of this work, which are then worked out in detail in the following sections. In Sec. 2 we review the formalism of quadratic Liouvillians and explain our random sampling. Spectral properties (spectral boundary, phase transition, and spectral gap) are discussed in Sec. 3 and steady-state properties (distribution, purity, and statistics) in Sec. 4. In Sec. 5, we present concluding remarks and possible further directions. Technical calculations and proofs are presented in a series of six appendices.

## 1.1  Main results

In the thermodynamic limit $N_F \to \infty$, the single-particle properties of random quadratic Liouvillians, specified by Eqs. (1) and (2), are determined by two parameters: the dissipation strength $g$ and the ratio of the number of jump operators to the number of fermions $m = M/(2N_F)$. In Fig. 1(a), we plot the phase diagram of this system in the $1/g$ versus $m$ plane. For large enough $m$ and small enough $g$, the system is in phase I, characterized by a single-body spectrum supported on a simply-connected region of the complex plane, see Fig. 1(b). When $g$ is increased or $m$ decreased across a critical value, a phase transition occurs and, in phase II, the single-body spectrum splits into two disconnected components, see Fig. 1(c). The existence of these two regions signals the existence of an intermediate period of metastability, during which an extensive number of modes coexist without (considerable) decay. The critical line separating the two phases [dashed line in Fig. 1(a)] and the boundaries of the single-body spectral support can be computed analytically [52–55].

For very strong dissipation $g \to \infty$, the transition occurs at $m = 1/2$ and corresponds to the decoupling of some fermionic degrees of freedom from the dynamics. Indeed, the Hamiltonian contribution vanishes when $g \to \infty$ and there is an insufficient number of jump operators ($M < 2N_F$) to couple all $2N_F$ fermionic creation and annihilation operators to the environment. Below the transition [$m < 1/2$, red line in Fig. 1(a)] the decoupled fermions exhibit nonergodic

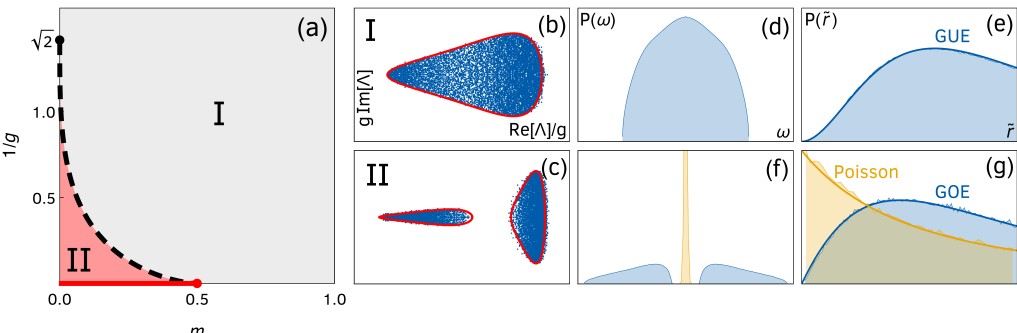

Figure 1: Schematic representation of the main results of this paper, the single-particle spectral and steady-state properties of random quadratic Liouvillians. (a) Phase diagram in the $1/g$ versus $m$ plane. Phases I and II are separated by the critical line $g_c(m)$ (dashed line). In the limit $g \to \infty$, the phase transition occurs at $m = 1/2$, with the spectral gap vanishing for $m < 1/2$ (red line). (b, c) Single-body spectrum in the complex plane, with a single connected component in phase I (b) and two disconnected components in phase II (c). The spectral boundary (red line) can be obtained analytically. (d, e) Steady-state properties for $g \to \infty$ and $m > 1/2$ (phase I). The steady-state has a single sector with the single-particle effective Hamiltonian eigenvalues, $\omega$, distributed according to RMT (d) and GUE statistics for the spacing ratio $\tilde{r}$ (e). (f, g) Steady-state properties for $g \to \infty$ and $m < 1/2$ (phase II, red line). The steady-state spectrum splits into an ergodic and a nonergodic sectors, in which the effective Hamiltonian's eigenvalues follow, respectively, RMT and a normal distribution (f). The $\tilde{r}$ statistics are Poisson in the nonergodic sector and interpolate from GUE to GOE in the ergodic sector as $m \to 0$ (g).

features, discussed in detail below. As $g$ is lowered to a finite value, the Hamiltonian starts to couple the dissipatively decoupled fermions and the critical value of $m$ decreases. At weak dissipation $g < 1/\sqrt{2}$, the Hamiltonian contribution is strong enough to couple all fermions and there is no transition.

The spectral gap, which sets the (inverse) timescale of relaxation to the steady-state, coincides for the single- and many-body spectra. It can also be obtained analytically and assumes a very simple form in the limits $g \to 0$ and $g \to \infty$. For weak dissipation, the spectral gap closes linearly with $g$ and $m$ (for all $m$), as expected from perturbation theory. On the other hand, for large dissipation, the spectral gap acts as an order parameter of the transition at $m = 1/2$. For $m < 1/2$, in the limit $g \to \infty$ the gap closes like $1/g$ leading to a gapless Liouvillian, signaling a slow approach to the steady-state. For $m > 1/2$, the gap has the linear scaling in $g$, typical of a dissipation-driven relaxation, growing with $m$ as $(\sqrt{2m} - 1)^2$, as dictated by the Marchenko-Pastur law. At the critical point $m = 1/2$, the gap closes as $(m/g)^{1/3}$. For large but finite $g$, the gap is nonzero for any value of $m$, but still exhibits qualitatively different behaviors above and below the transition.

The steady state, to which the system relaxes in the long-time limit, is Gaussian for quadratic Liouvillians and is thus fully characterized by its single-particle properties. In the limit $g \to 0$ (where there is no decoupling transition), the steady-state single-body spectrum is Gaussian, fully mixed, and displays Poisson statistics irrespectively of the value of $m$. As $g$ increases, there is a perturbative crossover well below the threshold $g = 1/\sqrt{2}$ for the appearance of the decoupling transition, and the steady state becomes distributed according to RMT [see Fig. 1(d)], is mixed but not fully mixed, and exhibits GUE statistics [see Fig. 1(e)]. The purity interpolates monotonously between the two limits $g \to 0$ and $g \to \infty$.

When $g$ is further increased, the steady state is also influenced by the decoupling transition. In the limit $g \to \infty$, the results can be obtained analytically through perturbation theory. Above the transition, $m > 1/2$, the properties attained after the perturbative crossover do not change. However, for $m < 1/2$, the spectrum of the steady state also splits into two independent sectors, see Fig. 1(f). The sector of the fermions coupled to the environment retains the properties of $m > 1/2$, except for the spectral statistics which, remarkably, crossover from GUE to GOE as $m \to 0$, see Fig. 1(g). The spectrum of the decoupled sector, on the other hand, is composed of uncorrelated Gaussian random variables displaying Poisson statistics (nonergodic behavior), see Fig. 1(g). These two sectors are well-separated for small-enough $m$, see Fig. 1(f), but overlap for larger $m$.

We emphasize that the nonergodic features of the steady state were obtained in the limit $g \to \infty$. However, they leave strong imprints in the dynamics at large but finite $g$ and $N_F$. Indeed, the interplay of the two sectors leads to a decrease of the purity, with a nonmonotonic behavior as a function of $g$, and to an interpolation between RMT and Poisson statistics.

Finally, we note that the properties of random quadratic Liouvillians above the transition are quantitatively similar to those identified in previous studies of fully random Liouvillians with unconstrained interactions [40, 41].[1] The nonergodic behavior below the transition is, however, not accessible to these fully-random Liouvillians.

## 2 Random quadratic Liouvillians

### 2.1 Liouvillian dynamics

The dynamics of the system can be entirely specified by looking at the Liouvillian eigenvalue problem. In fact, after determining a complete set of Liouvillian eigenmodes $\rho_j$ with eigenvalue $\Lambda_j$ [$\mathcal{L}(\rho_j) = \Lambda_j \rho_j$], we can write the evolution of any initial state $\rho(0) = \sum_j c_j \rho_j$ as[2]

$$\rho(t) = \sum_j c_j e^{\Lambda_j t} \rho_j. \tag{3}$$

All $\Lambda_j$ have a non-positive real part as ensured by the complete positivity of the Lindblad equation. Moreover, the Hermiticity preservation property guarantees that eigenvalues are real or come in complex-conjugated pairs. Generically, because of trace preservation, there is a single eigenstate, the steady state $\rho_{\text{NESS}}$, with corresponding zero eigenvalue. $\rho_{\text{NESS}}$ is invariant under time evolution, as

$$\frac{d\rho_{\text{NESS}}}{dt} = \mathcal{L}(\rho_{\text{NESS}}) = 0. \tag{4}$$

If there are no other eigenvalues with $\operatorname{Re}\Lambda_j = 0$, the system will relax to the steady state as $t \to \infty$, since all the other eigenmodes of the Liouvillian decay to zero. The rate at which the system relaxes to the steady state is dictated by the Liouvillian spectral gap,[3] defined as

$$\text{Gap} = \min_{\Lambda_j \neq 0} |\operatorname{Re}\Lambda_j|. \tag{5}$$

---

[1]With the important caveat that here we studied the single-particle properties, whereas for non-quadratic models many-body properties have to be considered. Nonetheless, unconstrained fully-random Liouvillians also exhibit a connected spectrum, the gap (which is the same in the single- and many-body case) has exactly the same scaling as here, and the steady state exhibits a nonergodic-to-ergodic crossover and a corresponding change in purity as a function of $g$.

[2]We assume a generic case where Liouvillian has no non-trivial Jordan blocks. For the most general treatment see, e.g., Ref. [9].

[3]Though this is certainly true for finite systems, for systems of infinite dimension, the eigenmodes with eigenfrequencies close to the gap can add up and lead to an algebraic relaxation to the steady state.

## 2.2 Vectorization and adjoint fermions

To study the eigenvalue problem of the quadratic Liouvillian superoperator, it is convenient to recast it as a matrix acting in an enlarged Hilbert space, a procedure known as vectorization. In the Fock space of $N_F$ fermions, pure states are represented by $2^{N_F}$-dimensional vectors and mixed states and operators by $2^{N_F} \times 2^{N_F}$ matrices. Alternatively, we can see mixed states and operators as $2^{2N_F}$-dimensional vectors over a tensor product of two copies of the fermionic Fock space, while superoperators are represented by $2^{2N_F} \times 2^{2N_F}$ matrices. More explicitly, if $|\alpha\rangle$ and $|\beta\rangle$ are basis elements in the fermionic Fock space and $\hat{O}_{1,2}$ Fock-space operators, we map

$$|\alpha\rangle \langle\beta| \to |\alpha\rangle \otimes \langle\beta|^T , \tag{6}$$

$$\hat{O}_1 |\alpha\rangle \langle\beta| \hat{O}_2^\dagger \to \left(\hat{O}_1 \otimes \hat{O}_2^{\dagger T}\right) |\alpha\rangle \otimes \langle\beta|^T . \tag{7}$$

Following this procedure, the Liouvillian [Eq. (1)] is mapped to:

$$\mathcal{L} = -i\left(\hat{H} \otimes \mathbb{1} - \mathbb{1} \otimes \hat{H}^T\right) + g \sum_{\mu=1}^{M} \left\{ 2\hat{L}_\mu \otimes \hat{L}_\mu^* - \hat{L}_\mu^\dagger \hat{L}_\mu \otimes \mathbb{1} - \mathbb{1} \otimes (\hat{L}_\mu^\dagger \hat{L}_\mu)^T \right\} . \tag{8}$$

It is possible to generalize the notion of creation and annihilation operators to this space while keeping the Liouvillian quadratic, resembling the Hamiltonian of a free theory [7]. To enforce the canonical anticommutation relations in the vectorized representation also, we define the vector

$$\tilde{a} = \Big[ c_1 \otimes e^{i\pi\mathcal{N}^T}, \ldots, c_{N_F} \otimes e^{i\pi\mathcal{N}^T}, c_1^\dagger \otimes e^{i\pi\mathcal{N}^T}, \ldots, c_{N_F}^\dagger \otimes e^{i\pi\mathcal{N}^T},$$
$$\mathbb{1} \otimes c_1^{\dagger T} e^{i\pi\mathcal{N}^T}, \ldots, \mathbb{1} \otimes c_{N_F}^{\dagger T} e^{i\pi\mathcal{N}^T}, \mathbb{1} \otimes e^{i\pi\mathcal{N}^T} c_1^T, \ldots, \mathbb{1} \otimes e^{i\pi\mathcal{N}^T} c_{N_F}^T \Big]^T , \tag{9}$$

where $\mathcal{N} = \sum_i c_i^\dagger c_i$ is the number operator, which satisfies $\{\tilde{a}_i, \tilde{a}_j^\dagger\} = \delta_{ij}$ as required. $\tilde{a}_i$ are known as adjoint fermions. An important feature of this vector is that it is particle-hole symmetric, i.e.,

$$\left(\tilde{a}^\dagger\right)^T = \tilde{S}\tilde{a}, \qquad \tilde{S} = \begin{bmatrix} S & 0 \\ 0 & S \end{bmatrix}, \qquad S = \begin{bmatrix} 0 & \mathbb{1} \\ \mathbb{1} & 0 \end{bmatrix} . \tag{10}$$

$S$ and $\tilde{S}$ implement the particle-hole in the original and vectorized spaces, respectively. This vectorization scheme is equivalent to third quantization [7]. For an explicit demonstration, we refer the reader to Appendix A.

## 2.3 Single-particle spectrum and diagonalization

Recall that, as mentioned in Sec. 1 [see Eq. (2)], the Liouvillian in Eq. (1) for a system of $N_F$ fermions is said to be quadratic if

$$\hat{H} = \frac{1}{2} \sum_{i,j=1}^{2N_F} C_i^\dagger H_{ij} C_j \qquad \text{and} \qquad \hat{L}_\mu = \sum_{j=1}^{2N_F} l_{\mu j} C_j .$$

The $2N_F \times 2N_F$ matrix $H$, the so-called single-particle Hamiltonian, is Hermitian and can always be chosen to satisfy particle-hole symmetry, $S H^T S = -H$. In turn, the dissipative contribution to the Liouvilian is determined by the $M \times 2N_F$ non-Hermitian (and, in general, complex) matrix $\{l_{\mu j}\}_{\mu=1,\ldots,M}^{j=1,\ldots,N_F}$ (recall that $M$ is the number of independent decay channels).

Next, we define the matrices

$$N_{jk} = \sum_\mu l_{\mu j} l_{\mu k}^* , \qquad N_S = S N^T S , \tag{11}$$

the particle-hole-symmetric and antisymmetric combinations

$$\Gamma = \frac{N + N_S}{2}, \qquad \Gamma_B = \frac{N - N_S}{2},$$  (12)

and the non-Hermitian single-particle effective Hamiltonian

$$K = H - ig\Gamma.$$  (13)

In Appendix B.1 we show that

$$\mathcal{L} = -\frac{i}{2}\tilde{a}^\dagger L \tilde{a} - \frac{i}{2}\mathrm{Tr}(K),$$  (14)

where the single-particle Liouvillian is given by

$$L = \begin{bmatrix} H - ig\Gamma_B & -igN_S SJ \\ -igJSN & -J(H + ig\Gamma_B)^T J \end{bmatrix},$$  (15)

with

$$J = \begin{bmatrix} \mathbb{1} & 0 \\ 0 & -\mathbb{1} \end{bmatrix}.$$  (16)

Note that the matrix $L$ also satisfies particle-hole symmetry, $\tilde{S}L^T\tilde{S} = -L$.

As shown in Appendix B.2, we can (almost) always find a change of basis that renders the Liouvillian diagonal,

$$\mathcal{L} = -\frac{i}{2}\sum_i^{2N_F} \beta_i b_i' b_i,$$  (17)

where $b_i$ and $b_i'$ (note, however, $b_i' \neq b_i^\dagger$) satisfy the canonical anticommutation relations

$$\{b_i, b_j\} = 0, \quad \{b_i', b_j'\} = 0, \quad \text{and} \quad \{b_i, b_j'\} = \delta_{ij},$$  (18)

and $\{\beta_j\}_{j\in\{1,\ldots,2N_F\}}$ constitutes the single-body spectrum of the Liouvillian. Moreover, the $\beta_i$ coincide with the eigenvalues of the non-Hermitian Hamiltonian $K$. It is clear that due to the form of Eq. (17), the many-body spectrum of the Liouvillian is completely determined by the single-body spectrum (take all possible sums of subsets of $\{-i\beta_j/2\}_j$), which implies that all its properties are encapsulated in the latter. We will thus focus on studying the features of the single-body spectrum in Sec. 3.1.

Since the Liouvillian is quadratic and the single-body spectrum is entirely contained in the left-half plane, the Liouvillian spectral gap corresponds to the gap of the single-body spectrum. Thus, the following definition holds:

$$\mathrm{Gap} = \frac{1}{2}\min|\mathrm{Im}\beta_i|.$$  (19)

## 2.4 Steady state

From the diagonal form (17), the steady state $\rho_{\mathrm{NESS}}$ is found to be the state annihilated by all $b_i$. For a quadratic Liouvillian, any initial Gaussian state will remain Gaussian under time evolution, which implies that the steady state must be Gaussian as well. We can thus describe the steady state entirely by its correlation matrix,[4]

$$\chi_{ij} = \mathrm{Tr}\left(\rho_{\mathrm{NESS}} C_i C_j^\dagger\right),$$  (20)

---

[4]Also referred to as *covariance matrix*.

which satisfies the particle-hole symmetry

$$S\chi^T S = \mathbb{1} - \chi. \tag{21}$$

Alternatively, we can define an effective thermal-like Hamiltonian $\hat{\Omega}$ as

$$\rho_{\text{NESS}} = \frac{1}{Z}e^{\hat{\Omega}}, \qquad Z = \text{Tr}\, e^{\hat{\Omega}}. \tag{22}$$

This parametrization is convenient as it automatically takes care of the normalization and positive-definiteness of the steady-state density matrix. Moreover, because the steady state is Gaussian, $\hat{\Omega}$ can be fully characterized by the single-particle matrix $\Omega$,

$$\rho_{\text{NESS}} = \frac{1}{Z}e^{\frac{1}{2}\sum_{ij}C_i^\dagger \Omega_{ij}C_j}, \tag{23}$$

with normalization

$$Z = \text{Tr}\left(e^{\frac{1}{2}\sum_{ij}C_i^\dagger \Omega_{ij}C_j}\right) = \sqrt{\det\left(1 + e^{-\Omega}\right)}, \tag{24}$$

which is related to the correlation matrix through the matrix relation:

$$\chi = \left(1 + e^{\Omega}\right)^{-1}. \tag{25}$$

Remarkably, the steady-state correlation matrix can also be entirely determined by the single-particle non-Hermitian Hamiltonian. Indeed, as shown in Appendix B.3, one can solve the steady-state Lyapunov equation for the correlation matrix:

$$\chi K^\dagger - K\chi = igN. \tag{26}$$

Alternatively, $\chi$ can be constructed more efficiently using the right and left eigenvectors of the single-particle matrix $L$, see Eq. (B.32) below and Ref. [7]. The equivalence of the two methods is established in Appendix B.3.

## 2.5 Majorana basis

At this point, we change to the Majorana basis, which is more convenient for our purposes. It is implemented by the unitary transformation

$$U = \frac{1}{\sqrt{2}}\begin{bmatrix} \mathbb{1} & \mathbb{1} \\ i\mathbb{1} & -i\mathbb{1} \end{bmatrix}. \tag{27}$$

From it, we define the Majorana operators

$$f = \{f_1, \ldots, f_{2N_F}\}^T = \sqrt{2}UC, \tag{28}$$

where $C$ is the Nambu vector defined after Eq. (2), satisfying $f_i^\dagger = f_i$ and the anticommutation relation $\{f_i, f_j\} = \delta_{ij}$.

In this basis, $K$ is transformed into a more suitable matrix, $K^{(U)}$,

$$K^{(U)} = H^{(U)} - ig\Gamma^{(U)} = UKU^\dagger, \tag{29}$$

where $H^{(U)}$ is an anti-symmetric matrix,

$$H^{(U)T} = U^*H^T U^T = USH^T SU^\dagger = -H^{(U)}, \tag{30}$$

and $\Gamma^{(U)}$ is a symmetric matrix,

$$\Gamma^{(U)} = U\frac{l^T l^* + Sl^\dagger lS}{2}U^\dagger = \frac{1}{2}\left(l^{(U)T}l^{(U)*} + l^{(U)\dagger}l^{(U)}\right) = \frac{1}{2}\left(N^{(U)} + N^{(U)T}\right), \tag{31}$$

with $N^{(U)} = l^{(U)T}l^{(U)*}$ and

$$l^{(U)} = lU^T. \tag{32}$$

## 2.6 Random sampling

The characterization of the Liouvillian's spectrum and steady state is now completely determined by the specification of matrices $H^{(U)}$ and $l^{(U)}$, which can only be obtained from the knowledge of the system's Hamiltonian and interactions with the environment. However, assuming that the dynamics is generic, one can argue based on Jayne's principle of maximal entropy that they are well described by a random matrix of a symmetry class consistent with the symmetries of the system. In this case, we restrict $H^{(U)}$ to the set of Hermitian matrices satisfying particle-hole symmetry $H^{(U)^T} = -H^{(U)}$ and randomize all other degrees of freedom, corresponding to $2N_F \times 2N_F$ Gaussian random matrices of class D in the Altland-Zirnbauer classification [56]. The simplest way to achieve this is to draw a matrix $H_{\text{int}}$ from the Ginibre orthogonal ensemble (GinOE) [57], i.e., sample a real matrix from the probability distribution $\sim \exp\{-N_F \operatorname{Tr}(H_{\text{int}}^\dagger H_{\text{int}})\}$ and then set

$$H^{(U)} = \frac{i}{2}\left(H_{\text{int}} - H_{\text{int}}^T\right).\tag{33}$$

On the other hand, since we do not impose any restriction on the jump operators, we sample $l^{(U)}$ from the Ginibre unitary ensemble (GinUE) [57] of rectangular $M \times 2N_F$ matrices, i.e., sample a complex matrix from the probability distribution $\sim \exp\left\{-N_F \operatorname{Tr}\left(l^{(U)^\dagger} l^{(U)}\right)\right\}$. This is equivalent to saying that the $2N_F \times 2N_F$ matrix $N^{(U)}$ is drawn from the complex Wishart ensemble [also known as the Laguerre unitary ensemble (LUE)].

Now that we have established a procedure to determine matrices $H^{(U)}$ and $l^{(U)}$, we can turn to the study of the spectral and steady-state properties of random Liouvillians, which are entirely determined by single-body ones, allowing us to focus just on the properties of the latter. We start with the spectral properties.

# 3 Spectral properties

We are interested in the support of the single-body spectrum, as it contains information on the relevant timescales of the problem. The rightmost boundary point gives the spectral gap (recall that the single- and many-particle gaps coincide). The width of the spectrum along the imaginary axis is related to the timescale for the oscillations of the states' phases. Finally, if the spectral support splits into several components there is a hierarchy of decay times [44,45], with separate sets of modes decaying at different rates, interspersed by periods of metastability [46].

We first show that our random model can be mapped exactly, in the limit $N_F \to \infty$, to a slightly different non-Hermitian Hamiltonian whose boundary has been computed using free probability and use it to identify a phase transition in the single-body spectrum, see Sec. 3.1. Then, in Sec. 3.2, we focus on the spectral gap, studying it in detail, both numerically and analytically, as a function of $g$ and $m$.

## 3.1 Single-body spectrum

### 3.1.1 Spectral boundary

As mentioned in Sec. 2 and proven in Appendix B.2, the spectrum of the single-body Liouvillian matrix $L$ coincides with that of $K = H - ig\Gamma$. In Refs. [52–55], the authors studied the spectrum of a related random matrix $H_{\text{eff}} = H_r - ig_r\Gamma_r$, with $H_r$ and $\Gamma_r = AA^T$ being $2N_F \times 2N_F$ Hermitian matrices drawn from the Gaussian Orthogonal Ensemble (GOE) and the real Wishart ensemble [also known as the Laguerre orthogonal ensemble (LOE)], respectively. ($A$ is a real $2N_F \times M$ matrix.) Using replicas [52], supersymmetry [53], diagrammatics [54],

or free probability [55], they established that, in the limit $N_F \to \infty$, the spectrum of $H_{\text{eff}}$ is supported on a bounded region in the complex plane delimited by a boundary that satisfies the equation

$$x^2 = -\frac{m}{g_r y} - \left(\frac{g_r}{1-2g_r y} + \frac{m}{2y} - \frac{1}{2g_r}\right)^2 , \qquad (34)$$

where $m = M/(2N_F)$ and $x + iy$ represents a point in the complex plane.[5]

In our case, the problem is slightly different as, to obtain the single-body spectrum, we need to calculate the eigenvalues of $K^{(U)} = H^{(U)} - ig\Gamma^{(U)}$ rather than $H_{\text{eff}} = H_r - ig_r\Gamma_r$. Quite remarkably, apart from numerical prefactors and subleading $1/N_F$ corrections, the eigenvalue distribution of $K^{(U)}$ and $H_{\text{eff}}$ coincide, as we argue below.

The difference between $H^{(U)}$ and $H_r$ stems from the fact that the former is Hermitian and anti-symmetric whereas the latter is Hermitian and symmetric. Although they belong to distinct symmetry classes, their resolvents and, hence, their eigenvalue distributions match to leading order in $1/N_F$, as we review in Appendix C. Because the resolvent is the only property of $H^{(U)}$ that enters the determination of the eigenvalue distribution of $K^{(U)}$, we can interchange it with $H_r$.

In addition, $\Gamma^{(U)}$ is drawn from a symmetrized complex Wishart ensemble, in contrast to $\Gamma_r$, which is drawn from the real Wishart ensemble. However, we can also simply draw $\Gamma^{(U)}$ from the real Wishart ensemble, provided we double the number of jump operators ($m = M/2N_F \to 2M/2N_F = 2m$):

$$
\begin{aligned}
\Gamma^{(U)}_{j,k} &= \frac{1}{2}\sum_{\mu=1}^{M}\left(\left(l^{(U)}\right)_{\mu,j}\left(l^{(U)}\right)^{*}_{\mu,k} + \left(l^{(U)}\right)_{\mu,k}\left(l^{(U)}\right)^{*}_{\mu,j}\right) \\
&= \sum_{\mu=1}^{M}\left(\left(l^{(U)}\right)^{R}_{\mu,j}\left(l^{(U)}\right)^{R}_{\mu,k} + \left(l^{(U)}\right)^{I}_{\mu,j}\left(l^{(U)}\right)^{I}_{\mu,k}\right) = \sum_{\mu=1}^{2M}\left(l^{(U)}_r\right)_{\mu,j}\left(l^{(U)}_r\right)_{\mu,k} ,
\end{aligned}
\qquad (35)
$$

where $\left(l^{(U)}\right)_{\mu,j} = \left(l^{(U)}\right)^{R}_{\mu,j} + i\left(l^{(U)}\right)^{I}_{\mu,j}$ and $l^{(U)}_r$ is a $2M \times 2N_F$ real matrix built from concatenating $\left(l^{(U)}\right)^{R}$ (taken as the first $M$ rows of $l^{(U)}_r$) and $\left(l^{(U)}\right)^{I}$ (the last $M$ rows).

With the equivalence of the spectra of $K^{(U)}$ and $H_{\text{eff}}$ established, to obtain the boundary of the single-body spectrum of the Liouvillian in the limit $N_F \to \infty$, we must simply replace $m$ by $2m$ in Eq. (34), together with $x$ by $-2y$ and $y$ by $2x$ (since the single-body spectrum is obtained from the spectrum of $K$ by multiplication by $-i/2$):

$$y^2 = -\frac{m}{4gx} - \left(\frac{g/2}{1-4gx} + \frac{m}{4x} - \frac{1}{4g}\right)^2 . \qquad (36)$$

In Fig. 2, we plot the curve parametrized by Eq. (36) in the complex plane and compare it with the single-body spectrum of $-iK/2$ obtained numerically by exact diagonalization (ED), for three different points in the parameter space $(m, g)$. We observe that it adjusts perfectly to the boundary of the spectrum in all cases. The very small number of outliers can be attributed to finite-size effects, since the boundary becomes sharp in the limit $N_F \to \infty$. In particular, one can show [52–55] that for $N_F \to \infty$, no states lie outside the boundary with probability going to 1.

All the spectral information, including the phase diagram in the $1/g$ versus $m$ plane and the spectral gap, can be extracted from Eq. (36), as we discuss in the remainder of this section.

---

[5]Note that Eq. (34) differs from that in Refs. [54] by some numerical factors, which have their origin in the different normalizations used. Using our conventions, $H_r$ is sampled from the probability distribution $\sim \exp\{-N_F \text{Tr}(H_r^2)\}$ and $A$ from the distribution $\sim \exp\{-N_F \text{Tr}(AA^T)\}$.

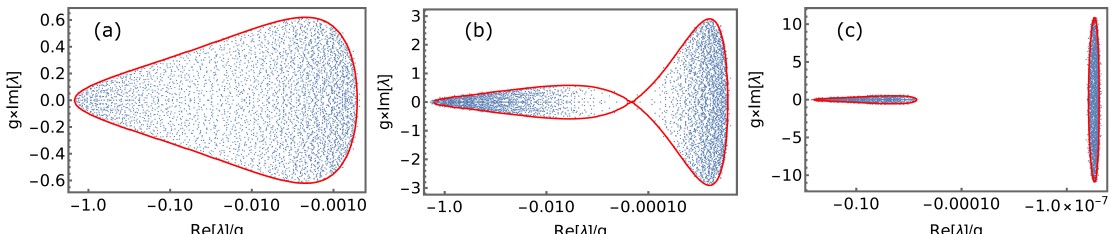

Figure 2: Single-body spectrum of the Liouvillian for $m = 0.2$ and $g = 1$ (a), $g \approx 5.2$ (b), and $g = 20$ (c). The $2N_F$ blue points are obtained numerically by exact diagonalization (for a single realization with $N_F = 4000$) and are bounded by the red curve, which is parametrized by Eq. (36).

### 3.1.2 Phase diagram

From Fig. 2, we can observe that two different behaviors of the single-body spectrum emerge for different values of $m$, for a given $g > 1/\sqrt{2}$. For large enough $m$ and small enough $g$, the single-body spectrum is supported on a simply connected region of the complex plane, see Fig. 2(a). We call this region of $(m, g)$ space phase I. When $g$ is increased or $m$ decreased across some critical value $g_c(m)$ or $m_c(g)$, a phase transition occurs [52–55], see Fig. 2(b), and the single-body spectrum splits into two disconnected components. In phase II, for large enough $m$ and small enough $g$, the two components of the single-body spectrum are well separated, see Fig. 2(c).

In the $1/g$ versus $m$ plane, phases I and II are separated by a critical line that can be obtained analytically from Eq. (36), see Fig. 1(a). Indeed, assuming the spectrum to be the union of convex sets in the complex plane (an assumption verified in all numerical simulations) and since the boundary described by Eq. (36) is clearly symmetric under reflections across the real axis, the number of components of the spectrum is determined by the number of real roots of the equation $y^2 = 0$, which can be rewritten as:

$$
\begin{aligned}
x^4 + \left( (1 + 2m)g - \frac{1}{2g} \right) x^3 + \left( \frac{1}{16g^2} + \frac{g^2(1 - 2m)^2}{4} - m - \frac{1}{4} \right) x^2 \\
+ \frac{mg}{4} \left( \frac{1}{2g^2} + 1 - 2m \right) x + \frac{m^2}{16} = 0 .
\end{aligned}
\tag{37}
$$

Since Eq. (37) is quartic in $x$, it can have at most four real roots. In that case, the spectrum of the Liouvillian is formed by two disconnected components, each bounded between two real roots of Eq. (37), i.e., the system is in phase II. Alternatively, there could be only two real roots (and a pair of complex-conjugated roots that do not contribute to the boundary of the spectrum), in which case the system would be in phase I. The phase transition between these two situations occurs when two real roots coalesce into a double root. The number of real roots of Eq. (37) is controlled by its discriminant

$$
\Delta_4 = -\frac{m^3}{128} \left[ \left( 1 - 2(1 - 2m)g^2 \right)^3 + 216mg^4 \right] .
\tag{38}
$$

As discussed, the phase transition corresponds to the merger of two distinct real roots into a double root, which occurs if and only if the discriminant vanishes. Setting $\Delta_4^2 = 0$ thus gives the critical line in the $1/g$ vs $m$ plane [52, 53, 55]:

$$
\frac{1}{g_c} = \sqrt{2 \left( 1 - (2m)^{1/3} \right)^3} .
\tag{39}
$$

If $\Delta_4 > 0$, i.e., $g > g_c$, then Eq. (37) has four real roots and we are in phase II. In the converse case $\Delta_4 < 0$, i.e., $g < g_c$, there are only two real roots and the spectrum belongs to phase I.

From Fig. 1(a) and Eq. (39), it is clear that a critical point exists at $(m, g) = (1/2, \infty)$. Below $m = 1/2$, there is always a finite value of $g$ for which the spectrum splits into two distinct regions. However, the closer $m$ gets to $1/2$, the larger are the values of $g$ required for the phase transition to occur, tending to infinity as $m \to 1/2$. Above $m = 1/2$, the spectrum is connected for all values of $g$ and just stretches indefinitely as we increase $g$. On the other hand, for $g < 1/\sqrt{2}$, the system also belongs to phase I for all values of $m$. As discussed in Sec. 1.1, below $g = 1/\sqrt{2}$ the Hamiltonian contribution to the Lindbladian is strong enough to couple all degrees of freedom, preventing the system from splitting into two decoupled components.

In phase II, when we increase $g$, the two regions drift further and further apart from each other, both becoming very thin stripes in the limit $g \to \infty$, one of them aligned with the imaginary axis with an increasingly small absolute real part and the other aligned with the real axis with an increasingly large absolute real part. Physically, the existence of these two distinct regions of eigenvalues means that the group of modes associated with the region with a larger absolute real part will decay much faster than the others. In an intermediate time window, the system will evolve to a metastable state in which only the modes that belong to the region with smaller absolute real parts are populated. Eventually, those also fade away and the system reaches the steady state.

## 3.2 Spectral gap

We now turn to the computation of the spectral gap, obtained from the single-body spectrum through Eq. (19). In the limit $N_F \to \infty$, the boundary of the single-body spectrum is determined by Eq. (36) and, thus, the gap is just the largest real root of Eq. (37). In Fig. 3(a), we plot the gap as a function of $g$ for three different values of $m$ obtained both numerically from ED and exactly from Eq. (37), showing perfect agreement between the two. While the expression for the gap can be obtained analytically for any $g$, its precise functional form is rather complicated and not particularly enlightening. In what follows we will see, however, that simple scaling expressions can be obtained in the limits of weak ($g \to 0$) and strong ($g \to \infty$). The latter case is particularly interesting due to the influence of the decoupling transition.

### 3.2.1 Weak dissipation

Regardless of the value of $m$, the gap goes to zero in the limit $g \to 0$. This behavior is expected as, for $g = 0$, there is just unitary evolution (the Liouvillian becomes simply the von Neumann generator) and so all the eigenvalues lie on the imaginary axis, which means that, by definition, the gap vanishes. In fact, since in Fig. 3(a) the slope of all curves approaches 1 for very small $g$, we see that, in this limit, Gap $\propto g$, a result that can be understood perturbatively. Since a Taylor expansion of the gap around $g = 0$ yields Gap $= \sum_{n=1}^{\infty} c_n g^n$, the mentioned scaling behaviour holds unless $c_1 = 0$. Now, $H^{(U)}$ is a random Hermitian and anti-symmetric matrix and, in the space of all such matrices, the set of degenerate matrices has measure zero. Thus, we can safely apply non-degenerate first-order perturbation theory and conclude that $c_1 = \min\left(\{v_i^\dagger \Gamma^{(U)} v_i\}_i\right)$, where $\{v_i\}_i$ is the set of normalized eigenvectors of $H^{(U)}$. $\Gamma^{(U)}$ is a positive semi-definite matrix and thus $c_1 = 0$ only if there is a $v_i$ that belongs to the nullspace of $\Gamma^{(U)}$. Clearly, Eq. (35) implies that, for $m < 1/2$, $\Gamma^{(U)}$ has a nullspace of dimension $2N_F - 2M = 2N_F(1 - 2m)$. However, except for the trivial case $m = 0$, this is always a set of measure zero and, thus, the probability that one of the $v_i$ belongs to the nullspace of $\Gamma^{(U)}$ is 0. For $m > 1/2$, the nullspace of $\Gamma$ is empty. As a consequence, $c_1 \neq 0$ and Gap $\propto g$, when $g \to 0$, for all $m$.

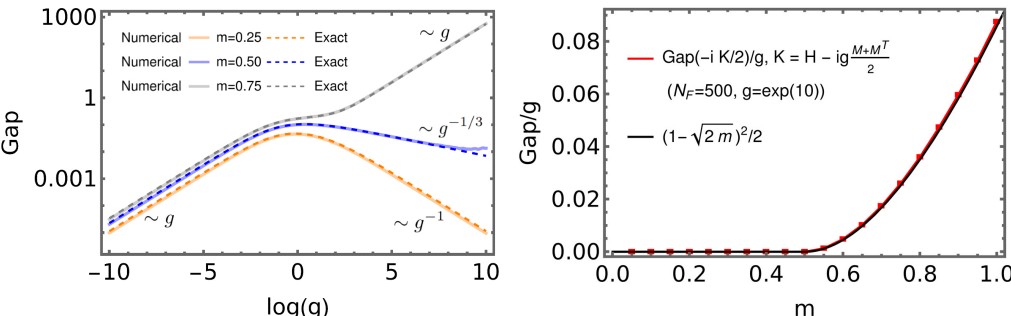

Figure 3: Average spectral gap as a function of the dissipation strength $g$ and number of jump operators $m$. Left: Gap as a function of $g$ for three different values of $m$: $m = 0.25$, 0.50, and 0.75. The full colored curves give the numerical results, the dashed black curve gives the exact result in the limit $N_F \to \infty$ obtained by solving Eq. (36). Right: Gap as a function of $m$ in the limit $g \to \infty$. The red dots correspond to the numerical results for $g = e^{10}$ and the black curve to the exact result of Eq. (40). In both panels, the numerical results were obtained by exact diagonalization, for $N_F = 500$ and 400 realizations, and match perfectly the analytical predictions.

### 3.2.2 Strong dissipation

On the other hand, the limit $g \to \infty$ has a nontrivial dependence on $m$ as depicted in Fig. 3(a). In fact, for $m < 1/2$, the spectral gap closes in this limit (Gap $\propto g^{-1}$), whereas for $m > 1/2$ it grows linearly with $g \to \infty$, which suggests a phase transition in the gap at $m = 1/2$, where an intermediate behaviour is observed, Gap $\propto g^{-1/3}$. The quantity $\lim_{g \to \infty} (\text{Gap}/g)$ can be used as an order parameter for the phase transition as it vanishes for $m \leq 1/2$ and acquires a nonzero finite value for $m > 1/2$. In Fig. 3(b), we plot it as a function of $m$ and compare it with the exact result

$$\lim_{g \to \infty} \frac{\text{Gap}}{g} = \frac{1}{2} \max \left\{ \left( \sqrt{2m} - 1 \right)^2, 0 \right\}, \tag{40}$$

that follows from the Marchenko-Pastur law [58]. Indeed, we have $\lim_{g \to \infty} iK^{(U)}/g = \Gamma^{(U)}$, where $\Gamma^{(U)}$ is a $2N_F \times 2N_F$ matrix drawn from a real Wishart ensemble with rank $2M$, and hence the gap coincides the hard edge of the Marchenko-Pastur distribution.

The critical behavior can be traced to the existence or not of zero eigenvalues in the spectrum of $\Gamma^{(U)}$ and understood perturbatively. Since $K^{(U)} = g \left( H^{(U)}/g - i\Gamma^{(U)} \right)$, we can expand the eigenvalues of $K^{(U)}/g$ in powers of $1/g$ and write the gap as Gap $= g \sum_{n=0}^{\infty} d_n g^{-n}$. To zeroth order, the spectrum of $K^{(U)}/g$ is the spectrum of $-i\Gamma^{(U)}$. If $m > 1/2$, $\Gamma^{(U)}$ is a positive-definite matrix and thus $d_0 = \min(\text{Im}(\{\gamma_i\}_i)) > 0$, where $\{\gamma_i\}_i$ is the spectrum of $\Gamma^{(U)}$. This justifies the linear growth of the spectral gap with $g$ for $m > 1/2$. However, for $m < 1/2$, some of the $\gamma_i$ vanish and, consequently, $d_0 = 0$. We must look at the next term in the expansion and since, to do that, we need to calculate first-order corrections to the zero eigenvalues of $\Gamma^{(U)}$, we must resort to degenerate perturbation theory. The corrections are thus given by the eigenvalues of $(1/g)w_i^\dagger H^{(U)} w_j$, where $\{w_j\}_j$ is an orthonormal basis of the nullspace of $\Gamma^{(U)}$. Since, however, $H^{(U)}$ is Hermitian, all the corrections to these eigenvalues are real and thus do not affect the gap, leading to $d_1 = 0$. Only at second order in $1/g$ do non-zero corrections to the gap arise, which means that Gap $= g \sum_{n=2}^{\infty} d_n g^{-n}$. In the large $g$ limit, Gap $\propto g^{-1}$ as confirmed by Fig. 3(a).

Since $m = 1/2$ is the critical point, the above expansion in powers of $1/g$ does not hold. We need, therefore, to resort to Eq. (36) to determine the gap's scaling behavior with $g$. In Appendix D we perform an asymptotic analysis of the solutions of Eq. (36) for large $g$ and

show that, at $m = 1/2$, Gap $\propto g^{-1/3}$. The same procedure can be employed as an alternative to the perturbation theory above, in order to obtain the scaling behavior of the gap for $m > 1/2$ and $m < 1/2$ (and the corresponding prefactors), which is also done in Appendix D.

### 3.2.3 Comparison with non-quadratic models

We conclude this section by comparing the results obtained here with the gap of a fully random (non-quadratic) Liouvillian [38–40]. Because the single-body gap coincides with the many-body gap the results of the two models can be directly compared. In the weak dissipation regime, the same linear growth with the dissipation parameter is found in both cases and it is the expected perturbative result. The strong dissipation regime is, as we have seen, richer. In the fully-random case, the role of the parameter $m$ is played not by the ratio of dissipation channels to the number of degrees of freedom ($2N_F$ here), but by the number of channels itself $m \to M/2$. The regime $m < 1/2$ is, therefore, inaccessible since $M$ is a positive integer. For fully-random Liouvillians with more than one decay channel (corresponding to $m > 1/2$) we also observed a linear-in-$g$ growth of the gap, which again is the expected perturbative result for the dissipation-dominated dynamics. The case of a single decay channel (corresponding to $m = 1/2$ here) shows the same closing of the gap with $\sim g^{-1/3}$ (once one accounts for different normalization conventions). This closing was interpreted as a Zeno-like phase, but it was not understood why it is not observed for more than one decay channel (corresponding to $m > 1/2$). We can now understand the difference between $m > 1/2$ and $m < 1/2$ as a transition in which some degrees of freedom become decoupled from the environment and, thus, protected from dissipation, with $m = 1/2$ corresponding to the critical transition point. Our findings thus shine a new light on the special role played by fully-random Liouvillians with a single jump operator (corresponding here to $m = 1/2$). Remarkably, the scaling with $m$ also coincides in both cases (see Appendix D for a computation of the prefactors, which can be compared with Ref. [40]) and we conclude that the spectral gap coincides in quadratic and fully random Liouvillians in the mutually accessible regimes ($m \geq 1/2$), pointing towards a high degree of universality in dissipative quantum chaos. On the other hand, realistic models with constrained interactions, such as quadratic Liouvillians, contain an additional regime ($m < 1/2$) of suppressed dissipation.

## 4 Steady-state properties

We now turn to the characterization of the steady state, to which the system relaxes in the long-time limit. Because of its Gaussian nature (see Sec. 2), we will base our discussion on the level of the single-particle correlation matrix $\chi$ or the thermal-like Hamiltonian $\Omega$, which are related through Eq. (25) (proven in Appendix B.3). We will start by studying the spectrum of these single-body operators, which describe the occupation probabilities of different single-particle states in the long-time limit, see Sec. 4.1. We will focus on the limits of very weak ($g \to 0$) and very strong dissipation ($g \to \infty$), which can be studied perturbatively. To infer the behavior of the steady state as a function of $g$, we will consider, in Sec. 4.2, the first nontrivial moment of the steady-state distribution—the purity—which captures its degree of mixing. Finally, in Sec. 4.3 we probe the ergodicity of the steady state by analyzing the single-particle level statistics of $\Omega$. As was the case for the single-body spectrum and spectral gap, the results are qualitatively different for $m > 1/2$ and $m < 1/2$.

## 4.1 Spectral distribution

The correlation matrix $\chi$ is the solution of Eq. (26), reproduced here for convenience:

$$\chi K^\dagger - K\chi = igN\,.$$

Because of the particle-hole symmetry of $\chi$, Eq. (21), its eigenvalues come in pairs $\{\lambda_i, 1-\lambda_i\}$, $i \in \{1,...,N_F\}$. Consequently, the spectrum of $\Omega$ (denoted as $\{\omega_i\}_i$) is formed by the pairs $\{\omega_i, -\omega_i\}$. The statistical behavior of $\lambda_i$ and $\omega_i$ depends on the value of the parameters $g$ and $m$. Equation (26) can be solved perturbatively for $g \to 0$ and $g \to \infty$, allowing us to analytically study the spectral distribution in these two limiting cases. The details of the perturbative expansion are given in Appendix E, while here we state the final results and work out the consequences for the single-particle effective Hamiltonian $\Omega$.

### 4.1.1 Weak dissipation

In the weak dissipation limit ($g \to 0$), the eigenvalues of $\chi$ to first order in $g$ are determined by Eq. (E.3) of Appendix E, which we rewrite in the Majorana basis by performing the unitary transformation $U$, Eq. (27):

$$\lambda_i = \frac{v_i^\dagger N^{(U)} v_i}{v_i^\dagger \left(N^{(U)} + \left(N^{(U)}\right)^T\right) v_i}\,, \tag{41}$$

with $v_i$ an eigenvector of $H^{(U)}$. Since $H^{(U)}$ is Hermitian and anti-symmetric, it can always be diagonalized by a unitary matrix of the form $\mathcal{O}U$, for some orthogonal matrix $\mathcal{O}$. Therefore, in the eigenbasis of $H^{(U)}$, Eq. (41) reads

$$\lambda_i = \frac{\left(N^{(H)}\right)_{i,i}}{\left(N^{(H)} + (U^T U)^\dagger \left(N^{(H)}\right)^T U^T U\right)_{i,i}} = \frac{\left(N^{(H)}\right)_{i,i}}{\left(N^{(H)} + \left(SN^{(H)}S\right)^T\right)_{i,i}}\,, \tag{42}$$

where $U^T U = S$, $N^{(H)} = \left(l^{(H)}\right)^T \left(l^{(H)}\right)^*$ and $\left(l^{(H)}\right) = l^{(U)}\mathcal{O}U^*$. From Eq. (42), we conclude that the spectrum of $\chi$ is composed of $N_F$ independent pairs $\{\lambda_i, 1-\lambda_i\}$, with

$$\lambda_i = \frac{\sum_{\mu=1}^M \left|\left(l^{(H)}\right)_{\mu,i}\right|^2}{\sum_\mu^M \left(\left|\left(l^{(H)}\right)_{\mu,i}\right|^2 + \left|\left(l^{(H)}\right)_{\mu,i+N_F}\right|^2\right)} = \left(1 + \frac{\sum_\mu^M \left|\left(l^{(H)}\right)_{\mu,i+N_F}\right|^2}{\sum_\mu^M \left|\left(l^{(H)}\right)_{\mu,i}\right|^2}\right)^{-1}\,. \tag{43}$$

The eigenvalues of $\Omega$, in turn, are:

$$\omega_i = \log\left(\frac{\sum_\mu^M \left|\left(l^{(H)}\right)_{\mu,i+N_F}\right|^2}{\sum_\mu^M \left|\left(l^{(H)}\right)_{\mu,i}\right|^2}\right)\,. \tag{44}$$

Each $\left|\left(l^{(H)}\right)_{\mu,2i}\right|^2 = \text{Re}\left(\left(l^{(H)}\right)_{\mu,i}\right)^2 + \text{Im}\left(\left(l^{(H)}\right)_{\mu,i}\right)^2$ is a sum of two random variables with average $\sigma^2 = 1$ and variance $2\sigma^4 = 2$ (recall that $\text{Re}\left(l^{(H)}\right)_{\mu,i}$ and $\text{Im}\left(l^{(H)}\right)_{\mu,i}$ are sampled from a normal distribution with zero mean and standard deviation $\sigma = 1$). We can now resort to the central limit theorem to argue that, for sufficiently large $M$,

$$\omega_i \approx \log\left(\frac{1 + \frac{1}{\sqrt{M}}Z'_{i+N_F}}{1 + \frac{1}{\sqrt{M}}Z'_i}\right) \approx \sqrt{\frac{2}{M}}Z_i\,, \tag{45}$$

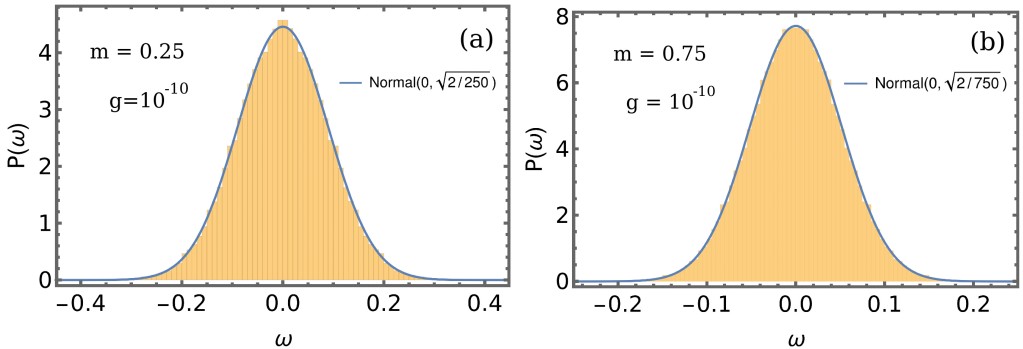

Figure 4: Spectrum of the effective Hamiltonian $\Omega$ for $N_F = 500$, $m = 0.25$ (a) and $m = 0.75$ (b), and $g = 10^{-10}$. The histograms were obtained by exact diagonalization for 100 disorder realizations, with the correlation matrix obtained by the method described in Appendix E. The blue line represents a normal distribution with variance $\sqrt{2/M}$, as predicted by Eq. (52).

where $Z'$ and $Z$ are, respectively, a set of $2N_F$ and $N_F$ random variables following a normal distribution with unit variance. We conclude that the steady-state single-body spectrum in the weak dissipation limit is composed of a set of uncorrelated Gaussian random variables with variance $2/M$. In Fig. 4, we plot this prediction against the numerical results obtained by exact diagonalization and find perfect agreement.

It is clear from Eq. (45) that $\lim_{M\to\infty} \chi = \mathbb{1}/2$, which means that, in the limits $N, M \to \infty$ with $m$ fixed and $g \to 0$, the steady state of a random quadratic Liouvillian is the fully-mixed state. We will elaborate on this in Sec. 4.2 below. Moreover, we expect single-body Poisson spectral statistics because of the uncorrelated nature of different $\omega_i$, a prediction confirmed in Sec. 4.3.

### 4.1.2 Strong Dissipation

In the strong dissipation limit ($g \to \infty$), $m$ plays a significant role in the statistical behavior of the spectrum of $\Omega$, with two qualitatively different regimes for $m > 1/2$ and $m < 1/2$, which can be traced back to the decoupling transition at $m = 1/2$.

If $m > 1/2$, $\chi$ is determined, in the eigenbasis of $\Gamma^{(U)}$, by Eq. (E.6) of Appendix E:

$$\chi_{i,j} = \frac{w_i^\dagger N^{(U)} w_j}{\gamma_i + \gamma_j}, \tag{46}$$

where $\{\gamma_i\}_i$ and $\{w_i\}_i$ are the set of eigenvalues and orthonormal eigenvectors of $\Gamma^{(U)}$, respectively. Since $\Gamma^{(U)}$ is positive semidefinite, $\gamma_i \geq 0$. Note that, for $m > 1/2$, there are no zero eigenvalues of $\Gamma^{(U)}$, and hence $\gamma_i + \gamma_j \neq 0$ always holds. From Eq. (46) it follows that the eigenvalues of $\chi$ are, in general, correlated. Since they are completely determined by the eigenvectors and eigenvalues of a Wishart matrix, we conjecture that they can be related to the Marchenko-Pastur law, although we were not able to do so explicitly.

When $m < 1/2$, $\chi$ cannot be obtained from Eq. (E.6) since $\gamma_i + \gamma_j = 0$ for some pairs $(i, j)$. More precisely, $\chi$ becomes a matrix that acts separately in two subspaces (see Appendix E): subspace $\bar{N}$ spanned by the eigenvectors of $\Gamma^{(U)}$ with a corresponding non-zero eigenvalue, $(w_{\bar{N}})_i$, and its complement, subspace $N$, which is also the nullspace of $\Gamma^{(U)}$, spanned by the eigenvectors $(w_N)_i$. Correspondingly, the spectrum splits into two independent components. Anticipating the results found below, we call $\bar{N}$ the RMT or ergodic sector and $N$ the Poisson or nonergodic sector.

The component of $\chi$ that acts in $\bar{N}$ (i.e., in the sector with no vanishing $\gamma_i$), $\chi_{\bar{N}\bar{N}}$, is directly obtained from Eq. (E.6) by replacing $w_i$ with $\left(w_{\bar{N}}\right)_i$. The only difference to the case $m > 1/2$ is thus a reduction of dimensionality. On the other hand, in Appendix E, we show that the eigenvalues of $\chi_{NN}$ (i.e., in the sector with zero eigenvalues of $\Gamma^{(U)}$) are given by Eq. (E.16), which can be rewritten as

$$\lambda_i = \frac{1}{2}\left(\mathbb{1} + \frac{(w_N)_i^\dagger H_{N\bar{N}}^{(U)}\left(\Gamma_{\bar{N}\bar{N}}^{(U)}\right)^{-1}\left(\Gamma_B^{(U)}\right)_{\bar{N}\bar{N}}\left(\Gamma_{\bar{N}\bar{N}}^{(U)}\right)^{-1}H_{\bar{N}N}^{(U)}(w_N)_i}{(w_N)_i^\dagger H_{N\bar{N}}^{(U)}\left(\Gamma_{\bar{N}\bar{N}}^{(U)}\right)^{-1}H_{\bar{N}N}^{(U)}(w_N)_i}\right), \tag{47}$$

where we used that $\left(\Gamma_{\bar{N}\bar{N}}^{(U)}\right)^T = \left(\Gamma_{\bar{N}\bar{N}}^{(U)}\right)$, $\left(H_{N\bar{N}}^{(U)}\right)^T = -\left(H_{\bar{N}N}^{(U)}\right)$ and $\Gamma_{\bar{N}\bar{N}}^{(U)} + \left(\Gamma_B^{(U)}\right)_{\bar{N}\bar{N}} = N_{\bar{N}\bar{N}}^{(U)}$. Given the similarity of Eqs. (41) and (47), it is possible to replicate the argument we used for the weak-dissipation case to find the eigenvalues $\lambda_i$ and $\omega_i$ in this sector. To do so, we first note that $H_{NN}^{(U)}$ is anti-symmetric and thus we can diagonalize it with the matrix $\mathcal{O}U$, for an orthogonal matrix $\mathcal{O}$. As a consequence, Eq. (47) can be rewritten as

$$\lambda_i = \frac{1}{2} + \frac{\left(H_{N\bar{N}}^{(H)}\left(\Gamma_{\bar{N}\bar{N}}^{(U)}\right)^{-1}\left(\Gamma_B^{(U)}\right)_{\bar{N}\bar{N}}\left(\Gamma_{\bar{N}\bar{N}}^{(U)}\right)^{-1}H_{\bar{N}N}^{(H)}\right)_{i,i}}{\left(H_{N\bar{N}}^{(H)}\left(\Gamma_{\bar{N}\bar{N}}^{(U)}\right)^{-1}H_{\bar{N}N}^{(H)}\right)_{i,i}}, \tag{48}$$

where $H_{N\bar{N}}^{(H)} = H_{N\bar{N}}^{(U)}\mathcal{O}U$. $H_{N\bar{N}}^{(U)}$ is a $4N_F m \times 2N_F(1-2m)$ matrix with random purely imaginary entries. As a consequence, the first $N_F(1-2m)$ columns of $H_{N\bar{N}}^{(H)}$ are independent complex-valued random vectors and $\left(H_{N\bar{N}}^{(H)}\right)_{j,k} = \left(H_{N\bar{N}}^{(H)}\right)_{j,k+N_F(1-2m)}$, for $k \in \{1, ..., N_F(1-2m)\}$. Just as in the weak dissipation case, this implies that the spectrum of $\chi$ associated with this sector is also composed of pairs $\{\lambda_i, 1-\lambda_i\}$. Due to this symmetry, from now on we will just focus on $i \in \{1, ..., N_F(1-2m)\}$ in Eq. (48).

To proceed, we assume that different unitary transformations were performed in the $\bar{N}$ sector in the numerator and in the denominator: in the former to the eigenbasis of $\left(\Gamma^{(U)}\right)^{-1}\Gamma_B^{(U)}\left(\Gamma^{(U)}\right)^{-1}$ and in the latter to the eigenbasis of $\Gamma_{\bar{N}\bar{N}}^{(U)}$. Note that since all these transformations are unitary in their respective subspaces and independent of $H_{N\bar{N}}^{(H)}$, the entries of $H_{N\bar{N}}^{(H)}$ remain independent and (complex) normally distributed after the transformation. Therefore, denoting the eigenvalues of $\left(\Gamma^{(U)}\right)^{-1}\Gamma_B^{(U)}\left(\Gamma^{(U)}\right)^{-1}$ by $\alpha_i$, we conclude that

$$\begin{aligned}\left(H_{N\bar{N}}^{(H)}\left(\Gamma_{\bar{N}\bar{N}}^{(U)}\right)^{-1}\left(\Gamma_B^{(U)}\right)_{\bar{N}\bar{N}}\left(\Gamma_{\bar{N}\bar{N}}^{(U)}\right)^{-1}H_{\bar{N}N}^{(H)}\right)_{i,i} &= \sum_{\mu=1}^{2M}\alpha_\mu\left|\left(\tilde{H}_{\bar{N}N}^{(H)}\right)_{\mu,i}\right|^2, \\ \left(H_{N\bar{N}}^{(H)}\left(\Gamma_{\bar{N}\bar{N}}^{(U)}\right)^{-1}H_{\bar{N}N}^{(H)}\right)_{i,i} &= \sum_{\mu=1}^{2M}\gamma_\mu^{-1}\left|\left(\tilde{H}_{\bar{N}N}^{(H)}\right)_{\mu,i}\right|^2.\end{aligned} \tag{49}$$

Since $\left|\left(H_{N\bar{N}}^{(U)}\right)_{i,\mu}\right|^2$ is a random variable with average $\sigma = 1$ and variance $2\sigma^2 = 2$, by applying the central limit theorem we conclude that

$$\begin{aligned}\sum_{\mu=1}^{2M}\alpha_\mu\left|\left(H_{\bar{N}N}^{(H)}\right)_{\mu,i}\right|^2 &\sim \sum_{\mu=1}^{2M}\alpha_\mu + X = X, & X &\sim \text{Normal}\left(0, 2\sum_{\mu=1}^{2M}\alpha_\mu^2\right), \\ \sum_{\mu=1}^{2M}\gamma_\mu^{-1}\left|\left(H_{\bar{N}N}^{(H)}\right)_{\mu,i}\right|^2 &\sim \sum_{\mu=1}^{2M}\gamma_\mu^{-1} + Y, & Y &\sim \text{Normal}\left(0, 2\sum_{\mu=1}^{2M}\gamma_\mu^{-2}\right).\end{aligned} \tag{50}$$

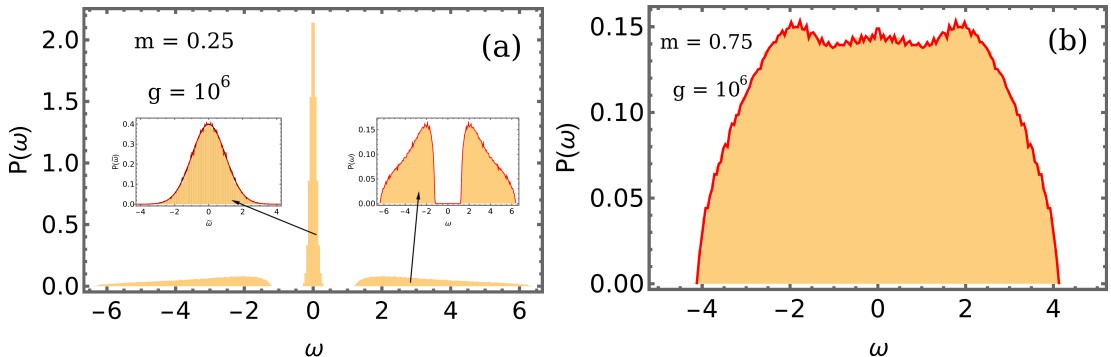

Figure 5: Spectrum of the effective Hamiltonian $\Omega$, for $N_F = 500$, $m = 0.25$ (a) and $m = 0.75$ (b), $g = 10^6$. The histograms were obtained by exact diagonalization for 100 disorder realizations, with the correlation matrix obtained by the method described in Appendix E. In (a), the spectrum splits into two independent sectors. The central region corresponds to the nonergodic sector (subspace $N$), and the distribution is Gaussian, as shown in the left inset, where we compare the eigenvalues $\omega$ centered at their mean and normalized by their standard deviation $[\tilde{\omega} = (\omega - \langle \omega \rangle)/\sigma_\omega]$ against a normal distribution with unit variance (red line). The remaining eigenvalues belong to the ergodic sector and are perfectly described by Eq. (46), see right inset (red line). In (b), there is no nonergodic sector, and all eigenvalues are distributed according to Eq. (46).

In fact, it is central to this argument that $\sum_{\mu=1}^{2M} \alpha_\mu = 0$, which is due to the fact that $\left(\Gamma^{(U)}\right)^{-1} \Gamma_B^{(U)} \left(\Gamma^{(U)}\right)^{-1}$ is anti-symmetric and thus $\mathrm{Tr}\left(\left(\Gamma^{(U)}\right)^{-1} \Gamma_B^{(U)} \left(\Gamma^{(U)}\right)^{-1}\right) = 0$.

Note that, despite hidden in the notation, the variables $X$ and $Y$ are not independent. However, it does not pose any problem as, in the thermodynamic limit, $\sum_{\mu=1}^{2M} \gamma_\mu^{-1} = 2M \langle \gamma^{-1} \rangle$ and therefore

$$\lambda_i \sim \frac{1}{2} + \frac{1}{2M} \frac{X}{\langle \gamma^{-1} \rangle + \frac{Y}{2M}} \sim \frac{1}{2} + \frac{1}{\sqrt{2M}} \mathrm{Normal}\left(0, 2\frac{\langle \alpha^2 \rangle}{\langle \gamma^{-1} \rangle^2}\right) + \mathcal{O}\left(\frac{1}{M}\right). \tag{51}$$

For $\omega_i$, this yields

$$\omega_i \sim \frac{4}{\sqrt{M}} \mathrm{Normal}\left(0, \frac{\mathrm{Var}(\alpha)}{\langle \gamma^{-1} \rangle^2}\right). \tag{52}$$

In summary, for $m > 1/2$, the eigenvalues of $\Omega$ are correlated according to RMT, as dictated by Eq. (46). In contrast, for $m < 1/2$, the steady-state spectrum splits into two sectors: in the first, the eigenvalues of $\Omega$ are still distributed according to Eq. (46), but in a space of smaller dimension; the second sector is formed by a set of uncorrelated Gaussian random variables (Poisson level statistics are checked in Sec. 4.3). As in the weak dissipation case, it becomes clear from Eq. (52) that $\lim_{M \to \infty} \chi_{NN} = \mathbb{1}/2$, signaling that the Poisson sector is fully mixed.

In Fig. 5, we plot the spectrum of the single-particle effective Hamiltonian obtained numerically from ED. In Fig. 5(a) (phase II, $m < 1/2$), the spectrum splits into two well-separated sectors. The central eigenvalues are normally distributed, see left inset, while the larger eigenvalues (in absolute value) follow Eq. (46), see right inset. The parameters $\langle \gamma^{-1} \rangle$ and $\mathrm{Var}(\alpha)$ appearing in the variance of the nonergodic eigenvalues can be written in terms of the matrices $\Gamma^{(U)}$ and $\Gamma_B^{(U)}$ as

$$\langle \gamma^{-1} \rangle = \mathrm{Tr}\left(\left(\Gamma_{\bar{N}\bar{N}}^{(U)}\right)^{-1}\right) \quad \text{and} \quad \mathrm{Var}(\alpha) = \mathrm{Tr}\left[\left(\left(\Gamma_{\bar{N}\bar{N}}^{(U)}\right)^{-1}\right)^2\right] + \mathrm{Tr}\left\{\left[\left(\left(\Gamma_{\bar{N}\bar{N}}^{(U)}\right)^{-1}\right)^2 (\Gamma_B^{(U)})_{\bar{N}\bar{N}}\right]^2\right\},$$

$$\tag{53}$$

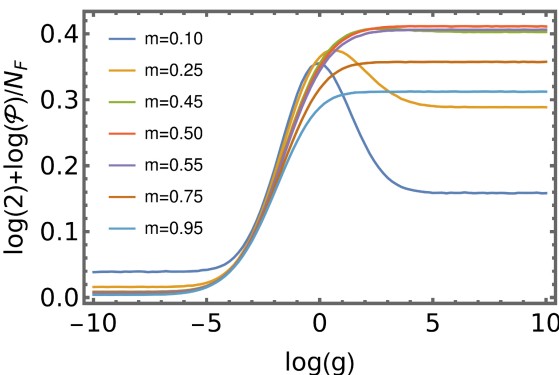

Figure 6: Reduced purity as a function of the dissipation strength $g$ for seven different values of $m = 0.10$–$0.95$, as given by Eq. (54). These results were obtained by exact diagonalization, with the correlation matrix obtained by the method described in Appendix E, for $N_F = 60$ and by averaging over 400 different realizations.

where averaging over the appropriate random ensemble is understood. While $\langle \gamma^{-1} \rangle$ and the first term in $\text{Var}(\alpha)$ can be re-expressed in terms of the Marchenko-Pastur distribution, we were unable to evaluate the second term in $\text{Var}(\alpha)$. While this prevents a parameter-free comparison with the numerical results, we still confirmed perfect Gaussianity of the nonergodic sector of the steady state in the inset of Fig. 5(a). In Fig. 5(b) (phase I, $m > 1/2$), all eigenvalues belong to a single ergodic sector.

## 4.2 Purity

To study the steady-state spectral distribution away from the limits $g \to 0, \infty$, we look at its moments as a function of $g$. The first moment is identically one, because of the normalization of probability. The purity, $\mathcal{P} = \text{Tr}\left(\rho_{\text{NESS}}^2\right)$, is the lowest nontrivial moment and quantifies the degree of mixing of the steady state.

Since it is possible to express the steady-state density matrix as a function of its correlation matrix $\chi$, the same applies to the purity:

$$\mathcal{P} = \text{Tr}\left(\rho_{\text{NESS}}^2\right) = \frac{\text{Tr}\left(e^{C^\dagger \Omega C}\right)}{\text{Tr}\left(e^{\frac{1}{2} C^\dagger \Omega C}\right)^2} = \frac{\det\left(1 + e^{-2\Omega}\right)^{\frac{1}{2}}}{\det\left(1 + e^{-\Omega}\right)} = \sqrt{\det\left((1-\chi)^2 + \chi^2\right)}, \qquad (54)$$

where we applied Eqs. (23)–(25) to obtain the third and fourth equalities.

In the following, instead of the purity itself, we consider the quantity

$$\frac{1}{N_F} \log \frac{\mathcal{P}}{\mathcal{P}_{\text{FM}}} = \log 2 + \frac{1}{N_F} \log \mathcal{P}, \qquad (55)$$

which we dub reduced purity, and where $\mathcal{P}_{\text{FM}} = 2^{-N_F}$ is the purity of the fully-mixed state. The reduced purity, which coincides with the shifted and rescaled second Rényi entropy, is finite in the limit $N_F \to \infty$ and its lower bound (corresponding to the fully-mixed state) is zero.

In Fig. 6, we plot the reduced purity as a function of $g$ for different values of $m$ and $N_F = 60$. In the limit $g \to 0$, it tends to a fixed value close to zero, for all $m$. In fact, it is expected to converge to zero in the limit $N_F \to \infty$ regardless of $m$, since, as we proved in the last subsection, $\lim_{M \to \infty} \chi = \mathbb{1}/2$ (note that this proof also justifies the slower convergence for smaller values of $m$ depicted in Fig. 6).

As we increase $g$, two different behaviors of the purity emerge independently of $N_F$ in the large-$N_F$ limit. If $m \geq 1/2$, it increases monotonically with $g$, stabilizing to a constant in

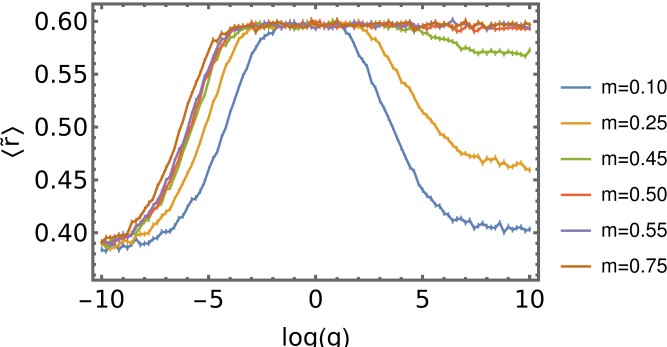

Figure 7: Mean level-spacing ratio $\langle \tilde{r} \rangle$ as a function of the dissipation strength $g$ for six different values of $m = 0.1$–$0.75$. These results were obtained by exact diagonalization, with the correlation matrix obtained by the method described in Appendix E, for $N_F = 60$ and by averaging over 400 different realizations.

the limit of very large $g$. The value of this plateau decreases with $m$. On the other hand, for $m < 1/2$, the purity initially increases, attaining a maximum at a finite value of $g$, and then it starts decreasing, converging in the limit $g \to \infty$ to a value smaller than that for $m > 1/2$, and which increases with $m$. The nonmonotonic behavior of the purity is a consequence of the splitting of the steady-state spectrum into two independent sectors at $m = 1/2$. In the RMT sector, the steady state has a finite reduced purity, which follows the same functional form as for $m > 1/2$. On the other hand, in the Poisson sector, the steady-state is fully mixed, as shown in the previous section, and hence has vanishing reduced purity. The competition between the two sectors determines the total purity of the steady state. Since the nullspace of $\Gamma$ has dimension $2N_F(1-2m)$, the Poisson sector becomes increasingly dominant as $m$ decreases. In the limit $m \to 0$, almost all eigenvalues of $\chi$ are obtained from the component $\chi_{NN}$ and we conclude that the reduced purity converges to zero.

To conclude this subsection, we note that even though the reduced purity completely determines the steady state when it is equal to 0 ($\chi = \mathbb{1}/2$), it provides only partial information when assuming other values. In particular, since it is not a proper measure of entanglement, it would be interesting to check whether the entanglement of the steady state also obeys a similar nonmonotonic behavior for $m < 0.5$. As the steady state is mixed, the entanglement entropy is also not a good measure of entanglement and one needs to resort to other more suitable quantities such as the mutual information or the negativity [59, 60]. However, this analysis falls out of the scope of this paper and we leave it for future work.

## 4.3 Spectral statistics

We conclude our study of the steady state by studying the single-particle spectral statistics of $\Omega$, which characterize the ergodicity, or lack thereof, of the steady state. We will focus on the distribution of consecutive spacing ratios, $\tilde{r}_n$ [61, 62]. Let $s_n$ be the sequence of differences between consecutive eigenvalues of $\Omega$. Then, $\tilde{r}_n$ is defined as [61]

$$\tilde{r}_n = \min\left(\frac{s_{n+1}}{s_n}, \frac{s_n}{s_{n+1}}\right). \tag{56}$$

This quantity has the advantage of being independent of the local level density and bounded ($0 < \tilde{r}_n < 1$). If the steady state is ergodic, the spectral correlations of $\Omega$ coincide with those of a random matrix of the appropriate symmetry class. On the other hand, if it is nonergodic, it will display Poisson statistics characteristic of uncorrelated random variables. The spacing ratio distributions for all three classes of level repulsion—the Gaussian Unitary Ensemble (GUE),

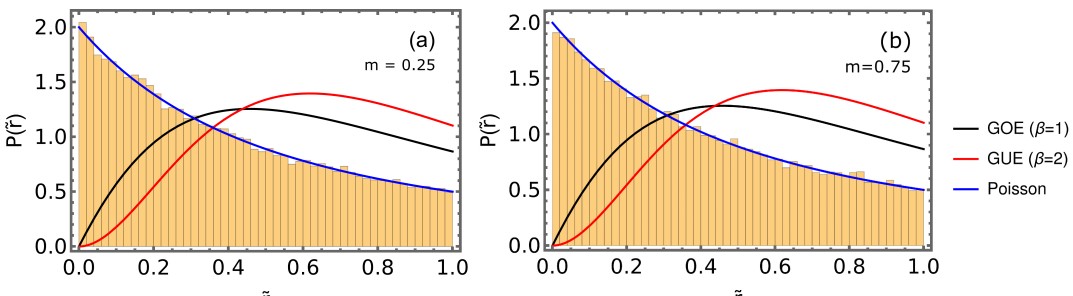

Figure 8: Level spacing ratio distribution $P(\tilde{r})$ for $N_F = 500$, $m = 0.25$ (a) and $m = 0.75$ (b), in the limit $g \to 0$. The histograms were obtained by exact diagonalization, with the correlation matrix obtained by Eq. (41), for 100 different realizations. The colored lines correspond to the analytical results of Eqs. (57) and (58). We find perfect agreement with the Poisson prediction, confirming nonergodic behavior.

Gaussian Orthogonal Ensemble (GOE), and Gaussian Symplectic Ensemble (GSE)—are well known and approximated by [62]

$$P_\beta(\tilde{r}) = \frac{2}{Z_\beta} \frac{(\tilde{r} + \tilde{r}^2)^\beta}{(1 + \tilde{r} + \tilde{r}^2)^{3\beta/2+1}} \,, \tag{57}$$

with $Z_1 = 8/27$ ($\beta = 1$, GOE), $Z_2 = 81\sqrt{3}/4\pi$ ($\beta = 2$, GUE), and $Z_4 = 729\sqrt{3}/4\pi$ ($\beta = 4$, GSE), while for Poisson statistics it is

$$P_{\text{Poi}}(\tilde{r}) = \frac{2}{(1 + \tilde{r})^2} \,. \tag{58}$$

The average spacing ratio $\langle r \rangle = \int_0^1 d\tilde{r}\, P(\tilde{r})$ has become a popular measure of ergodicity and of the presence of time reversal in the ergodic phase. Its value for the GOE, GUE, and Poisson statistics is given by, respectively [62]:

$$\langle \tilde{r} \rangle_1 = 4 - 2\sqrt{3} \approx 0.536 \,, \quad \langle \tilde{r} \rangle_2 = \frac{2\sqrt{3}}{\pi} - \frac{1}{2} \approx 0.603 \,, \quad \text{and} \quad \langle \tilde{r} \rangle_{\text{Poi}} = 2 \log 2 - 1 \approx 0.386 \,. \tag{59}$$

(In the following, the GSE will play no role and will not be referred to further.)

As a first indication of the influence of $m$ and $g$ on the spectral statistics of $\Omega$, we plot $\langle \tilde{r} \rangle$ as a function of $g$ for different values of $m$ in Fig. 7. We observe that, in the limit $g \to 0$, $\langle \tilde{r} \rangle$ converges to $\langle \tilde{r} \rangle_{\text{Poi}}$ (the Poisson value) for all $m$. Figure 8 corroborates this result by clearly showing that the full distribution of $\tilde{r}$ in the limit $g \to 0$ perfectly matches $P_{\text{Poi}}(\tilde{r})$. In fact, this result follows from Eq. (45), where we showed that the spectrum of $\Omega$ is Gaussian and composed of $N_F$ uncorrelated pairs of eigenvalues.

On the other hand, in the limit $g \to \infty$, two distinct behaviors are observed in Fig. 7: if $m > 1/2$, $\tilde{r}$ converges to $\langle \tilde{r} \rangle_2$ (the GUE value), whereas, if $m < 1/2$, it converges to different values depending on $m$. The GUE ratio statistics for $m > 1/2$ are confirmed in Fig. 9(c). However, the results for $m < 1/2$ show a crossover from GUE statistics at $m = 1/2$ to Poisson statistics as $m \to 0$. Again, this behavior can be understood from our results in Sec. 4.1. There, we showed that, in the Poisson sector ($N$), the spectrum of $\Omega_{NN}$ is composed of uncorrelated pairs of eigenvalues [Eq. (52)]. This prediction is confirmed in Fig. 9(b). In contrast, in the ergodic sector ($\bar{N}$), the eigenvalues of $\Omega_{\bar{N}\bar{N}}$ are in general correlated. In our generic setting, $\chi$ (and hence $\Omega$) has no symmetries besides particle-hole symmetry and, consequently, we expect GUE statistics in the ergodic sector, see Fig. 9(b). Remarkably, in the $m \to 0$ limit, the ergodic

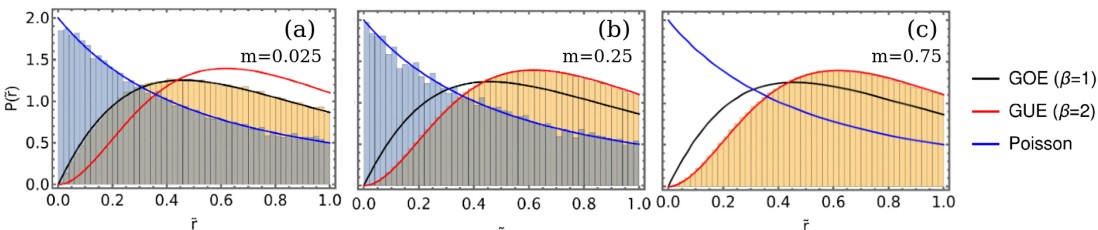

Figure 9: Level spacing ratio distribution $P(\tilde{r})$ in both sectors (RMT and Poisson) for $m = 0.025$ (a), $m = 0.25$ (b) and $m = 0.75$ (c), in the limit $g \to \infty$. In panel (c), there is just one histogram, because for $m > 0.5$ only the RMT sector is present. The histograms related to the RMT sectors (in yellow) were obtained by exact diagonalization of Eq. (46) for $N_F = 3000$ and 1000 different realizations. The histograms related to the Poisson sector (in blue) were obtained by solving Eq. 47) for $N_F = 500$ and 100 distinct realizations. The colored lines correspond to the analytical results of Eqs. (57) and (58). We find perfect agreement with the theoretical predictions in each case.

sector supports GOE statistics instead, see Fig. 9(a), a point we elaborate on further at the end of this section. Moreover, the relative weight of the ergodic and Poisson sectors changes as a function of $m$, although their dimensions always add up to $2N_F$. As a consequence, the spectra of $\Omega_{NN}$ and $\Omega_{\bar{N}\bar{N}}$ coexist in the interval $m \in (0; 1/2)$ and it is the variation of their relative contributions that causes the crossover from GUE to Poisson statistics as $m$ decreases. In the limit $m \to 0$, the dimension of $\bar{N}$ converges to $2N_F$ and the spectral statistics of $\Omega$ becomes Poissonian.

These two limits (weak and strong dissipation) have implications in the spectral statistics of the steady state at finite $g$. As depicted in Fig. 7, as $g$ increases from $-\infty$, the spectral correlations of the steady state exhibit a perturbative crossover from Poisson to GUE statistics for all $m$. A plateau is reached when $g$ is of the order of the inverse mean level spacing of the Hamiltonian. The plateau extends to infinity in the case $m > 1/2$, in agreement with the previous discussion of the limit $g \to \infty$. On the other hand, for $m < 1/2$, the plateau is of finite length, and, for sufficiently large $g$, the effects of the nullspace of $\Gamma$ become relevant. Therefore, the curve $\langle \tilde{r} \rangle$ decreases again to meet its expected value in the limit $g \to \infty$, that is, the result of the overlap of the two independent spectra previously mentioned.

Having established the main features of the dependence of the spectral correlations of the steady state on $g$ and $m$, we now discuss a curious aspect of the strong dissipation limit, namely, that the ergodic sector displays GOE statistics in the limit $m \to 0$, contrarily to the expectation of GUE statistics discussed above. As shown in Fig. 9(a) and (b), for finite $N_F$, there is a GUE-to-GOE crossover as $m$ decreases from $m = 1/2$ to $m \to 0$. In the thermodynamic limit, GUE statistics are attained for any finite $m$, while GOE statistics hold only in the strict limit $m \to 0$.[6] In Appendix F, we show that the spectrum of $\chi_{\bar{N}\bar{N}}$ can be expanded in powers of $\sqrt{m}$ and, to lowest nontrivial order ($\sim m$), it coincides with the spectrum of a matrix from class CI [56], which has the same level correlation as the GOE. We thus conclude that inside the phase $m < 1/2$ the spectral correlations of the steady state gradually change with $m$ due to two competing phenomena: an RMT-to-Poisson crossover due to the relative weight of the ergodic and nonergodic sectors and a GUE-to-GOE crossover in the ergodic sector.

---

[6]Note that $m \to 0$ corresponds to any value of $M$ that is either fixed or grows slower than $N_F$. The limit $m \to 0$ therefore still covers a wide range of physical systems.

# 5 Conclusions and Outlook

In this paper, we studied the single-body spectral and steady-state properties of a fermionic random quadratic Liouvillian. We studied the spectral support and boundary of the single-body Liouvillian spectrum, the spectral gap ruling the approach to the steady state, and the single-body distribution, purity, and single-body level statistics of the steady state. Our analysis focused on the phase transition observed in the single-body spectrum and its repercussions in the steady-state properties. More precisely, in phase I, the spectral and steady-state properties of quadratic and fully-random Liouvillians are qualitatively similar: the spectrum is formed by a single connected component; the gap grows linearly with dissipation strength; the steady-state purity is monotonic with dissipation strength; and there is a nonergodic-to-ergodic crossover in the steady-state level statistics, the full steady state being ergodic for sufficiently strong dissipation. In phase II, there are qualitative differences: the single-body spectrum splits into two disconnected components at a finite system-environment coupling strength; the spectral gap closes for strong dissipation; the purity is non-monotonic with dissipation strength; and the steady-state decouples into an ergodic and a nonergodic sectors.

In summary, our work identifies a regime of universal random Markovian dissipation but also illustrates the possibility of nonergodic behavior in quadratic open quantum systems (see also Ref. [63]) and the potential to suppress dissipation even in the presence of strong system-environment coupling. A natural extension of this work is to ask whether these nonergodic features survive interactions. Their robustness could be addressed, for instance, with the SYK Lindbladian [47–49]. Moreover, it is also not clear whether the nonergodic features of the steady state survive in the thermodynamic limit $N_F \to \infty$ at large but finite $g$ (i.e., whether the red line in Fig. 1(a) is smoothly connected to the rest of the diagram). This interesting question would require a more detailed finite-size scaling analysis and is deferred to future work. Other interesting possibilities are to consider bosonic Liouvillians and non-Markovian generators. Finally, further work is needed to determine whether the properties of stochastic Markovian dissipative models that we described are also present in more realistic models of open quantum systems.

## Acknowledgements

We thank Sergey Denisov and Kristian Wold for very fruitful discussions.

**Funding information**   This work was supported by Fundação para a Ciência e a Tecnologia (FCT-Portugal) through grants No. SFRH/BD/147477/2019 (LS) and UID/CTM/04540/2019 (PR). TP acknowledges ERC Advanced grant 694544-OMNES and ARRS research program P1-0402. This project was funded within the QuantERA II Programme that has received funding from the European Union's Horizon 2020 research and innovation programme under Grant Agreement No 101017733. ADL acknowledges support by the ANR JCJC grant ANR-21-CE47-0003 (TamEnt).

## A   Vectorization and third quantization

In this appendix, we explicitly show the equivalence between our vectorization scheme and the more standard third quantization, first introduced in Ref. [7].

In Ref. [7], it was noted that an orthonormal basis under the Hilbert-Schmidt inner product $\langle O_1 | O_2 \rangle = 2^{-N_f} \operatorname{Tr}\left(O_1^\dagger O_2\right)$ can be defined on the (vectorized) space of operators by considering

the set of all possible vectors

$$\left| P_{\underline{\alpha}} \right\rangle = \left| f_1^{\alpha_1} f_2^{\alpha_2} ... f_{2N_F}^{\alpha_{2N_F}} \right\rangle, \tag{A.1}$$

where $\alpha_i \in \{0, 1\}$ and $\{f_i\}_i$ represents a set of $2N_F$ Majorana fermions satisfying the Clifford algebra $\{f_j, f_k\} = 2\delta_{j,k}$. They can be obtained from the creation and annihilation operators, $c_j^\dagger$ and $c_j$, through

$$f_i = \sqrt{2} \sum_j U_{ij} C_j, \tag{A.2}$$

where $U$ is defined in Eq. (27).

At this point, creation and annihilation operators can be naturally defined on the vectorized space through

$$
\begin{aligned}
d_j \left| P_{\underline{\alpha}} \right\rangle &= \delta_{\alpha_j,1} \left| f_j P_{\underline{\alpha}} \right\rangle = \frac{1}{2} \left( \left| f_j P_{\underline{\alpha}} \right\rangle - \left| \Pi P_{\underline{\alpha}} \Pi f_j \right\rangle \right), \\
d_j^\dagger \left| P_{\underline{\alpha}} \right\rangle &= \delta_{\alpha_j,0} \left| f_j P_{\underline{\alpha}} \right\rangle = \frac{1}{2} \left( \left| f_j P_{\underline{\alpha}} \right\rangle + \left| \Pi P_{\underline{\alpha}} \Pi f_j \right\rangle \right),
\end{aligned}
\tag{A.3}
$$

where $\Pi = e^{i\pi\mathcal{N}}$ is the fermionic parity. The Liouvillian becomes quadratic when written as a function of these creation and annihilation operators and its diagonalization proceeds in a very similar manner to our case (see Appendix B).

Our goal is to find a linear transformation from the elements of $\tilde{a}$ to $d_j$ and $d_j^\dagger$. In order to do this, we first note that

$$
\begin{aligned}
(\Pi \otimes \mathbb{1}) d_j \left| P_{\underline{\alpha}} \right\rangle &= \begin{cases} -\frac{1}{2} \left( \left| f_j P_{\underline{\alpha}} \Pi \right\rangle + \left| P_{\underline{\alpha}} \Pi f_j \right\rangle \right) = \sum_{k=1}^{4N_F} Q_{jk}^{(1)} \tilde{a}_k \left| P_{\underline{\alpha}} \right\rangle, & \left( \Pi \otimes \Pi^T \right) \left| P_{\underline{\alpha}} \right\rangle = \left| P_{\underline{\alpha}} \right\rangle, \\ \frac{1}{2} \left( \left| f_j P_{\underline{\alpha}} \Pi \right\rangle - \left| P_{\underline{\alpha}} \Pi f_j \right\rangle \right) = -\sum_{k=1}^{4N_F} Q_{jk}^{(2)} \tilde{a}_k \left| P_{\underline{\alpha}} \right\rangle, & \left( \Pi \otimes \Pi^T \right) \left| P_{\underline{\alpha}} \right\rangle = -\left| P_{\underline{\alpha}} \right\rangle, \end{cases} \\
(\Pi \otimes \mathbb{1}) d_j^\dagger \left| P_{\underline{\alpha}} \right\rangle &= \begin{cases} -\frac{1}{2} \left( \left| f_j P_{\underline{\alpha}} \Pi \right\rangle - \left| P_{\underline{\alpha}} \Pi f_j \right\rangle \right) = \sum_{k=1}^{4N_F} Q_{jk}^{(2)} \tilde{a}_k \left| P_{\underline{\alpha}} \right\rangle, & \left( \Pi \otimes \Pi^T \right) \left| P_{\underline{\alpha}} \right\rangle = \left| P_{\underline{\alpha}} \right\rangle, \\ \frac{1}{2} \left( \left| f_j P_{\underline{\alpha}} \Pi \right\rangle + \left| P_{\underline{\alpha}} \Pi f_j \right\rangle \right) = -\sum_{k=1}^{4N_F} Q_{jk}^{(1)} \tilde{a}_k \left| P_{\underline{\alpha}} \right\rangle, & \left( \Pi \otimes \Pi^T \right) \left| P_{\underline{\alpha}} \right\rangle = -\left| P_{\underline{\alpha}} \right\rangle, \end{cases}
\end{aligned}
\tag{A.4}
$$

where

$$Q^{(1)} = -\frac{1}{\sqrt{2}} \begin{bmatrix} U & USJ \end{bmatrix} \quad \text{and} \quad Q^{(2)} = -\frac{1}{\sqrt{2}} \begin{bmatrix} U & -USJ \end{bmatrix}. \tag{A.5}$$

It is now easy to see that, after building the matrices

$$
\begin{aligned}
Q_1 &= \begin{bmatrix} Q^{(1)} \\ Q^{(2)} \end{bmatrix} = -\frac{1}{\sqrt{2}} \begin{bmatrix} U & 0 \\ 0 & U \end{bmatrix} \begin{bmatrix} \mathbb{1} & SJ \\ \mathbb{1} & -SJ \end{bmatrix}, \\
Q_{-1} &= -\begin{bmatrix} Q^{(2)} \\ Q^{(1)} \end{bmatrix} = \frac{1}{\sqrt{2}} \begin{bmatrix} U & 0 \\ 0 & U \end{bmatrix} \begin{bmatrix} \mathbb{1} & -SJ \\ \mathbb{1} & SJ \end{bmatrix},
\end{aligned}
\tag{A.6}
$$

we can relate $\tilde{a}$ with $D = \{d_1, \dots, d_{2N_F}, d_1^\dagger, \dots, d_{2N_F}^\dagger\}^T$ through

$$
\begin{cases} Q_1^\dagger (\Pi \otimes \mathbb{1}) D = \tilde{a}, & \text{if } \left( \Pi \otimes \Pi^T \right) \left| P_{\underline{\alpha}} \right\rangle = \left| P_{\underline{\alpha}} \right\rangle, \\ Q_{-1}^\dagger (\Pi \otimes \mathbb{1}) D = \tilde{a}, & \text{if } \left( \Pi \otimes \Pi^T \right) \left| P_{\underline{\alpha}} \right\rangle = -\left| P_{\underline{\alpha}} \right\rangle. \end{cases}
\tag{A.7}
$$

The two subspaces that appear in the previous equation correspond to the two parity sectors of Ref. [7]. In fact, if $\left( \Pi \otimes \Pi^T \right) \left| P_{\underline{\alpha}} \right\rangle = \left| P_{\underline{\alpha}} \right\rangle$ (respectively, $\left( \Pi \otimes \Pi^T \right) \left| P_{\underline{\alpha}} \right\rangle = -\left| P_{\underline{\alpha}} \right\rangle$), $P_{\underline{\alpha}}$ contains an even (respectively, odd) number of Majorana fermions.

We can now apply this transformation to Eq. (15) to reproduce the results in Ref. [7]:

$$\mathcal{L} = -\frac{i}{2}\tilde{a}^\dagger L\tilde{a} - \frac{i}{2}\text{Tr}(K) = \begin{cases} -\frac{i}{2}\tilde{D}^\dagger L_D^{(1)}\tilde{D} - \frac{i}{2}\text{Tr}(K), & \text{if } \left(\Pi \otimes \Pi^T\right)\left|P_{\underline{\alpha}}\right\rangle = \left|P_{\underline{\alpha}}\right\rangle, \\ -\frac{i}{2}\tilde{D}^\dagger L_D^{(-1)}\tilde{D} - \frac{i}{2}\text{Tr}(K), & \text{if } \left(\Pi \otimes \Pi^T\right)\left|P_{\underline{\alpha}}\right\rangle = -\left|P_{\underline{\alpha}}\right\rangle, \end{cases} \tag{A.8}$$

where

$$L_D^{(\gamma)} = Q_\gamma L Q_\gamma^\dagger = \begin{bmatrix} H^{(U)} - \frac{i}{2}\left(N^{(U)} + \left(N^{(U)}\right)^T\right) & -i\delta_{\gamma,1}\left(N^{(U)} - \left(N^{(U)}\right)^T\right) \\ -i\delta_{\gamma,-1}\left(N^{(U)} - \left(N^{(U)}\right)^T\right) & H^{(U)} + \frac{i}{2}\left(N^{(U)} + \left(N^{(U)}\right)^T\right) \end{bmatrix}, \tag{A.9}$$

and $\gamma \in \{-1, 1\}$ is an eigenvalue of $\Pi \otimes \Pi^T$. The matrices $H^{(U)}$ (Hermitian and anti-symmetric) and $N^{(U)}$ (Hermitian) of the previous expression are defined in Sec. 2.

It is now clear that Eqs. (A.8) and (A.9) in the even-parity sector ($\gamma = 1$) yield the same result as Eq. (22) of Ref. [7].[7] We have thus successfully established the equivalence between both paths to vectorization.

## B  Spectrum and steady state of fermionic quadratic Liouvillians

In this appendix, we prove the statements made in Sec. 2 about the spectrum and steady-state of quadratic fermionic Liouvillians. In Appendix B.1 we show how to write the single-body matrix of the Liouvillian using our vectorization scheme, proving Eq. (15). Then, in Appendix B.2 we show how to diagonalize this single-body matrix and, thus, prove Eq. (17). Finally, in Appendix B.3, we connect the steady-state and the single-body spectrum, proving Eq. (26).

### B.1  Single-body Liouvillian

In this section, we start by showing how to arrive at Eq. (15) from the vectorization of the Lindblad equation, given in Eqs (6)–(9). First, the von Neumann generator,

$$-i\left[\hat{H}, \rho\right] = -\frac{i}{2}\sum_{ij} H_{i,j}\left(C_i^\dagger C_j \rho - \rho C_i^\dagger C_j\right), \tag{B.1}$$

becomes, after vectorization,

$$-\frac{i}{2}\sum_{ij} H_{i,j}\left(C_i^\dagger C_j \otimes \mathbb{1} - \mathbb{1} \otimes C_j^T C_i^{\dagger T}\right)|\rho\rangle$$

$$= -\frac{i}{2}\sum_{ij} H_{i,j}\left(C_i^\dagger C_j \otimes \mathbb{1} + \mathbb{1} \otimes C_i^{\dagger T} C_j^T\right)|\rho\rangle \tag{B.2}$$

$$= -\frac{i}{2}\tilde{a}^\dagger \begin{bmatrix} H & 0 \\ 0 & -JH^T J \end{bmatrix}\tilde{a}|\rho\rangle.$$

Particle-hole symmetry, $SH^T S = -H$, was implicitly used in the first equality.

---

[7]Note that these equations seem to differ by some factors, but they are just a consequence of different definitions of the Lindblad equation (a factor of 2 in the dissipation part) and the fact that $\hat{H} = \frac{1}{2}\sum_{ij} C_i^\dagger H_{ij} C_j = \frac{1}{4}\sum_{ij}\left(H^{(U)}\right)_{ij} w_i w_j$.

One can proceed in a similar way to determine the vectorized version of the jump term in the Lindblad equation:

$$g \sum_{\mu} \hat{L}_{\mu} \rho \hat{L}_{\mu}^{\dagger} = g \sum_{\mu i j} l_{\mu,i}^* l_{\mu,j} C_i \rho C_j^{\dagger} \rightarrow \frac{1}{2} \tilde{a}^{\dagger} \begin{bmatrix} 0 & -g N_S S J \\ -g J S N & 0 \end{bmatrix} \tilde{a} |\rho\rangle . \tag{B.3}$$

Finally, the dissipative contribution,

$$-g \sum_{\mu} \frac{1}{2} \left\{ \rho, \hat{L}_{\mu}^{\dagger} \hat{L}_{\mu} \right\} = -g \sum_{\mu i j} \frac{1}{2} l_{\mu,i} l_{\mu,j}^* \left( C_i^{\dagger} C_j \rho + \rho C_i^{\dagger} C_j \right) , \tag{B.4}$$

becomes

$$\left( -\frac{1}{2} \tilde{a}^{\dagger} \begin{bmatrix} g\Gamma_B & 0 \\ 0 & g J \Gamma_B^T J \end{bmatrix} \tilde{a} - \frac{i}{2} \mathrm{Tr}(K) \right) |\rho\rangle , \tag{B.5}$$

where we used $\mathrm{Tr}(H) = \mathrm{Tr}(SHS) = -\mathrm{Tr}(H) = 0$.

Note that similarly to the case of $H$, we can always choose the adjoint fermionic creation and annihilation operators to satisfy particle-hole symmetry, which, in the vectorized space, reads as $\tilde{S} A^T \tilde{S} = -A$, where $\tilde{S}$ is defined in Eq. (10).

Summing Eqs. (B.2), (B.3), and (B.5), we can finally write the Liouvillian single-particle matrix:

$$\mathcal{L} = -\frac{i}{2} \tilde{a}^{\dagger} \begin{bmatrix} H - i g \Gamma_B & -i g N_S S J \\ -i g J S N & -J (H + i g \Gamma_B)^T J \end{bmatrix} \tilde{a} - \frac{i}{2} \mathrm{Tr}(K) , \tag{B.6}$$

concluding the proof of Eq. (15).

## B.2 Single-body spectrum

To diagonalize the Liouvillian, we must look at transformations of the form $\tilde{a} \rightarrow \tilde{b} = U \tilde{a} U^{-1}$, so that the canonical anticommutation relations are preserved. Let us consider the matrix $\exp\{-\frac{i}{2} \sum_{ij} \tilde{a}_i^{\dagger} \Delta_{ij} \tilde{a}_j\}$ with $\Delta = -\tilde{S} \Delta^T \tilde{S}$ (note that $\Delta$ can always be chosen to satisfy particle-hole symmetry). It follows that

$$\sum_{ij} \left[ \frac{1}{2} \tilde{a}_i^{\dagger} \Delta_{ij} \tilde{a}_j, \tilde{a}_k^{\dagger} \right] = \sum_i \tilde{a}_i^{\dagger} \Delta_{ik} , \tag{B.7}$$

which implies that

$$e^{\frac{i}{2} \sum_{ij} \tilde{a}_i^{\dagger} \Delta_{ij} \tilde{a}_j} \tilde{a}^{\dagger} e^{-\frac{i}{2} \sum_{ij} \tilde{a}_i^{\dagger} \Delta_{ij} \tilde{a}_j} = \tilde{a}^{\dagger} e^{i\Delta} . \tag{B.8}$$

Defining $\tilde{b} = \mathcal{R}^{-1} \tilde{a}$ and $\tilde{b}'^T = \tilde{a}^{\dagger} \mathcal{R}$, where $\mathcal{R} = e^{i\Delta}$, we find:

$$\mathcal{L} = -\frac{i}{2} \tilde{b}'^T \left( \mathcal{R}^{-1} L \mathcal{R} \right) \tilde{b} - \frac{i}{2} \mathrm{Tr}(K) . \tag{B.9}$$

We have reduced the diagonalization of the Liouvillian to the determination of the matrix $e^{-i\Delta}$ that diagonalizes $L$. Note that in general $\tilde{b}'^T \neq \tilde{b}^{\dagger}$, all we know is that $\{\tilde{b}_i, \tilde{b}'_j\} = \delta_{ij}$. Also, the particle-hole symmetry of $\Delta$ leads to the following restriction for the choice of $e^{-i\Delta}$:

$$\tilde{S} \mathcal{R}^T \tilde{S} = \tilde{S} e^{i\Delta^T} \tilde{S} = e^{-i\Delta} = \mathcal{R}^{-1} . \tag{B.10}$$

Any $\mathcal{R}$ that diagonalizes $L$ can be made to satisfy Eq. (B.10) by reordering columns, if necessary. To see this, suppose first that $v_L$ is a left eigenvector of $L$, i.e., $L^T v_L = \tau_\nu v_L$. Then,

$$L \tilde{S} v_L = -\tilde{S} L^T v_L = -\tau_\nu \tilde{S} v_L . \tag{B.11}$$

If $\{\tau^{(i)}\}_i$ is a set of $2N_F$ eigenvalues of $L$ with, for example, non-positive imaginary part and $\{v_L^{(i)}\}_i$ and $\{v_R^{(i)}\}_i$ are the corresponding left and right eigenvectors, respectively, then one can easily check that the matrix

$$\mathcal{R} = \mathcal{R}_1 \mathcal{P}_{23}, \tag{B.12}$$

with

$$\mathcal{R}_1 = \left[ v_R^{(1)}, \ldots, v_R^{(2N)}, \tilde{S} v_L^{(1)}, \ldots, \tilde{S} v_L^{(2N)} \right], \tag{B.13}$$

and

$$\mathcal{P}_{23} = \begin{bmatrix} \mathbb{1} & 0 & 0 & 0 \\ 0 & 0 & \mathbb{1} & 0 \\ 0 & \mathbb{1} & 0 & 0 \\ 0 & 0 & 0 & \mathbb{1} \end{bmatrix}, \tag{B.14}$$

does indeed satisfy Eq. (B.10).

At this point, all that is left to complete the diagonalization of the Liouvillian is to determine the matrix $\mathcal{R}_1$ explicitly. To this end, we first note that $L$ can be made block upper triangular after application of the transformation $U = \frac{1}{\sqrt{2}} \begin{bmatrix} \mathbb{1} & -\mathbb{1} \\ JS & JS \end{bmatrix}$:

$$U^{-1} L U = \begin{bmatrix} K & 2ig\Gamma_B \\ 0 & K^\dagger \end{bmatrix}. \tag{B.15}$$

Note that the previous equation implies that the eigenvalues of $L$ and $K$ coincide. It is straightforward to see that the matrix $L$ can be finally diagonalized by application of another change of basis, implemented by

$$V = \begin{bmatrix} R & X R^{\dagger^{-1}} \\ 0 & R^{\dagger^{-1}} \end{bmatrix}, \tag{B.16}$$

where $X$ is the solution of

$$X K^\dagger - K X = 2ig\Gamma_B. \tag{B.17}$$

Indeed,

$$(UV)^{-1} L (UV) = \begin{bmatrix} R^{-1} K R & 0 \\ 0 & \left( R^{-1} K R \right)^\dagger \end{bmatrix} = \begin{bmatrix} D_K & 0 \\ 0 & D_K^* \end{bmatrix}. \tag{B.18}$$

Recalling that $K = H - ig\Gamma$, with $\Gamma$ a positive semidefinite matrix, all the eigenvalues of $K$ have a non-positive imaginary part, which implies that

$$\left[ v_R^{(1)}, \ldots, v_R^{(2N_F)} \right] = \frac{1}{\sqrt{2}} \begin{bmatrix} R \\ JSR \end{bmatrix}. \tag{B.19}$$

To calculate the corresponding left eigenvectors we note that, after choosing the correct normalization, $X_L X_R = \mathbb{1}$, where $X_L$ ($X_R$) is a matrix whose rows (columns) are the left (right) eigenvectors of $L$. Thus, inverting $UV$ and keeping the first $2N_F$ rows gives us the desired result:

$$\left[ v_L^{(1)}, \ldots, v_L^{(2N_F)} \right] = \frac{1}{\sqrt{2}} \begin{bmatrix} \left( 1 + X^T \right) R^{T^{-1}} \\ JS \left( \mathbb{1} - X^T \right) R^{T^{-1}} \end{bmatrix}, \tag{B.20}$$

and, therefore,

$$\mathcal{R}_1 = \frac{1}{\sqrt{2}} \begin{bmatrix} R & S \left( 1 + X^T \right) R^{T^{-1}} \\ JSR & -J \left( \mathbb{1} - X^T \right) R^{T^{-1}} \end{bmatrix}. \tag{B.21}$$

Finally, the Liouvillian assumes the form

$$\mathcal{L} = -\frac{i}{2} \left( \tilde{b}'^T D_L \tilde{b} \right) - \frac{i}{2} \operatorname{Tr}(K), \tag{B.22}$$

where $D_L = \mathcal{R}^{-1} L \mathcal{R}$ obeying the particle-hole symmetry:

$$\tilde{S} D_L \tilde{S} = \tilde{S} \mathcal{R}^T L^T \left( \mathcal{R}^{-1} \right)^T \tilde{S} = -\mathcal{R}^{-1} L \mathcal{R} = -D_L \,. \tag{B.23}$$

Therefore, we can write $\text{Diagonal}(D_L) = \{\beta^{(1)}, -\beta^{(1)}, \beta^{(2)}, -\beta^{(2)}\}$, where $\beta^{(1)}$ and $\beta^{(2)}$ are both vectors of $N_F$ distinct eigenvalues of $K$. Actually, Eq. (B.22) can be further simplified by noting that it follows from Eq. (B.10) that

$$\tilde{S} \tilde{b} = \tilde{S} \mathcal{R}^{-1} \tilde{a} = \left( \tilde{a}^\dagger \mathcal{R} \right)^T = \tilde{b}' \,, \tag{B.24}$$

which means that $\tilde{b} = \{b^{(1)}, b'^{(1)}, b^{(2)}, b'^{(2)}\}$ and $\tilde{b}' = \{b'^{(1)}, b^{(1)}, b'^{(2)}, b^{(2)}\}$, where $b^{(1)}$ and $b^{(2)}$ are both vectors of $N_F$ annihilation operators and $b'^{(1)}$ and $b'^{(2)}$ of the corresponding creation operators. Equation (B.22) can be finally reduced to

$$
\begin{aligned}
\mathcal{L} &= -\frac{i}{2} \sum_{k=1}^{2} \sum_{j=1}^{N_F} \beta_j^{(k)} \left( b_j'^{(k)} b_j^{(k)} - b_j^{(k)} b_j'^{(k)} \right) - \frac{i}{2} \text{Tr}(K) \\
&= -\frac{i}{2} \sum_{k=1}^{2} \sum_{j=1}^{N_F} \beta_j^{(k)} b_j'^{(k)} b_j^{(k)} + \frac{i}{2} \left( \sum_{k=1}^{2} \sum_{j=1}^{N} \beta_j^{(k)} - \text{Tr}(K) \right) \\
&= -\frac{i}{2} \sum_{j=1}^{2N_F} \beta_j b_j' b_j \,,
\end{aligned}
\tag{B.25}
$$

where $b = \{b^{(1)}, b^{(2)}\}$ and $\beta = \{\beta^{(1)}, \beta^{(2)}\}$. The levels $-i\beta_j/2$ are the single-particle eigenvalues of the Liouvillian. The many-body eigenvalues are immediately obtained as

$$\Lambda_n = -\frac{i}{2} \sum_j n_j \beta_j \,, \tag{B.26}$$

where $n = \{n_1, \dots, n_{2N_F}\}$ and $n_j = 0, 1$.

## B.3 Steady state

From the discussion above, all elements of $\beta$ have nonpositive imaginary part, which implies that if $\text{Im}(\beta_i) \neq 0$ holds for all $\beta_i$, then the steady state, which satisfies $\mathcal{L}(\rho_{\text{NESS}}) = 0$, is unique and it is annihilated by all $b_i$. To see this, we note that $\mathcal{L}^\dagger(\mathbb{1}) = 0$ or, in vectorized notation, $\langle \mathbb{1} | \mathcal{L} = 0$. Thus, $\langle \mathbb{1} |$ and $|\rho_{\text{NESS}}\rangle$ form a biorthogonal left-right eigenvector pair. Using the anticommutation relations of $b_i$ and Eq. (B.25), this implies that

$$\langle \mathbb{1} | b_i' \mathcal{L} | O \rangle = - \langle \mathbb{1} | \mathcal{L} b_i' | O \rangle + \frac{i}{2} \beta_i \langle \mathbb{1} | b_i' | O \rangle \,, \tag{B.27}$$

or equivalently,

$$\langle \mathbb{1} | b_i' \left( \mathcal{L} - \frac{i}{2} \beta_i \right) | O \rangle = 0 \,, \tag{B.28}$$

for all vectorized operators $O$. Since the single-body spectrum of the Liouvillian is contained in the lower-half plane and, by assumption $\text{Im}(\beta_i) \neq 0$, we have $\mathcal{L} - \frac{i}{2}\beta_i \neq 0$ and, hence, Eq. (B.28) implies that, for all $i$, $\langle \mathbb{1} | b_i' = 0$ and, consequently, $b_i | \rho_{\text{NESS}} \rangle = 0$.

For quadratic systems, the steady state is Gaussian and completely determined by its $2N_F \times 2N_F$ correlation matrix,

$$\chi_{ij} = \text{Tr} \left( \rho_{\text{NESS}} C_i C_j^\dagger \right) = \sum_{k=1}^{2N} S_{jk} \langle \mathbb{1} | \tilde{a}_i \tilde{a}_k | \rho_{\text{NESS}} \rangle \,, \tag{B.29}$$

where $i \in \{1, ..., 2N_F\}$. Remarkably, $\chi_{ij}$ is also easily determined in terms of the single-body matrices $K$ and $\Gamma_B$. Indeed, we can write $\tilde{a}$ as a combination of $b$ and $b'$

$$\chi_{ij} = \sum_{k=1}^{2N_F} \sum_{\alpha,\beta=1}^{4N_F} S_{jk} \mathcal{R}_{i\alpha} \mathcal{R}_{k\beta} \langle \mathbb{1} | \, \tilde{b}_\alpha \tilde{b}_\beta \, | \rho_{\text{NESS}} \rangle \,, \tag{B.30}$$

and use the relations

$$\langle \mathbb{1} | \, b_i b_j \, | \rho_{\text{NESS}} \rangle = \langle \mathbb{1} | \, b'_i b'_j \, | \rho_{\text{NESS}} \rangle = \langle \mathbb{1} | \, b'_i b_j \, | \rho_{\text{NESS}} \rangle = 0 \quad \text{and} \quad \langle \mathbb{1} | \, b_i b'_j \, | \rho_{\text{NESS}} \rangle = \delta_{ij} \tag{B.31}$$

along with Eq. (B.21) to simplify Eq. (B.30):

$$\chi_{ij} = \sum_{k,n=1}^{2N} S_{jk} \mathcal{R}_{in}^{(1)} \mathcal{R}_{kn}^{(2)} = \frac{1}{2} (1 + X)_{ij} \,, \tag{B.32}$$

with $\mathcal{R}^{(1)}, \ldots, \mathcal{R}^{(4)}$ particle-hole blocks of the matrix $\mathcal{R}_1$, i.e., $\mathcal{R}_1 = \begin{bmatrix} \mathcal{R}^{(1)} & \mathcal{R}^{(2)} \\ \mathcal{R}^{(3)} & \mathcal{R}^{(4)} \end{bmatrix}$, and $X$ defined through Eq. (B.17). The single-particle matrix $X$ thus completely determines the steady state.

# C Resolvent of antisymmetric Hermitian random matrices

In this appendix, we show that the resolvent (also known as the Green's function) of anti-symmetric random Gaussian matrices (class D) coincides with that of GUE and GOE matrices (classes A and AI, respectively), in the large-$N_F$ limit. We will use the method of moments, in which we find the exact leading-order behavior of the $p$th moment of a matrix $Q$ and then infer the resolvent through the relation

$$G(z) = \frac{1}{N_F} \text{Tr} \left\langle \frac{1}{z - Q} \right\rangle = \frac{1}{N_F} \sum_{p=0}^{\infty} \frac{1}{z^{p+1}} \langle \text{Tr} \, Q^p \rangle \,. \tag{C.1}$$

Let us first compute the resolvent of a GUE matrix $Q$, with probability distribution $P(Q) \sim \exp\{-N_F/2 \, \text{Tr} \, M^2\}$. All odd moments vanish. All even moments can be related to the second moment (the propagator),

$$\langle Q_{ab} Q_{cd} \rangle = \frac{1}{N_F} \delta_{ad} \delta_{bc} \,, \tag{C.2}$$

through Wick's theorem, i.e., by summing over all possible pair contractions of the indices $a$, $b$, $c$, etc. For instance, the second moment is trivially

$$\langle \text{Tr} \, Q^2 \rangle = \sum_{a,b} \langle Q_{ab} Q_{ba} \rangle = N_F \,, \tag{C.3}$$

while the first nontrivial moment, the fourth, is given by:

$$\begin{aligned} \langle \text{Tr} \, Q^4 \rangle &= \sum_{a,b,c,d} \langle Q_{ab} Q_{bc} Q_{cd} Q_{da} \rangle \\ &= \sum_{a,b,c,d} (\langle Q_{ab} Q_{bc} \rangle \langle Q_{cd} Q_{da} \rangle + \langle Q_{ab} Q_{da} \rangle \langle Q_{bc} Q_{cd} \rangle + \langle Q_{ab} Q_{cd} \rangle \langle Q_{bc} Q_{da} \rangle) \\ &= N_F \left( 2 + \frac{1}{N_F^2} \right) \,. \end{aligned} \tag{C.4}$$

We can associate each Wick contraction with a perfect matching of the $p$ matrices $Q$ in the trace. Schematically, for $p = 4$:

$$\left\langle \mathrm{Tr}\, Q^4 \right\rangle = \left\langle \mathrm{Tr}\, QQQQ \right\rangle = \left\langle \mathrm{Tr}\, \overset{\frown}{Q}\overset{\frown}{Q}\overset{\frown}{Q}\overset{\frown}{Q} \right\rangle + \left\langle \mathrm{Tr}\, \overset{\frown}{Q}\overset{\frown}{QQ}\overset{\frown}{Q} \right\rangle + \left\langle \mathrm{Tr}\, \overset{\frown}{Q}\overset{\frown}{Q}\overset{\frown}{Q}\overset{\frown}{Q} \right\rangle .$$

Then, non-crossing (or planar) perfect matchings (e.g., the ones corresponding to the first two contractions in the fourth moment) contribute with a factor $N_F$ to the trace, while each crossing in the perfect matching suppresses the contribution of that contraction by $1/N_F^2$ (e.g., the third contraction in the fourth moments has a single crossing). The number of non-crossing perfect matchings of $2p$ elements is well-known to be the $p$th Catalan number, $C_p$, and, hence, we conclude that $\left\langle \mathrm{Tr}\, Q^{2p} \right\rangle = N_F C_p$ in the large-$N_F$ limit. Equation (C.1) can then be resummed.

Let us now turn to symmetric and antisymmetric random matrices, $Q_\pm = (Q \pm Q^T)/\sqrt{2}$. $Q_+$ is a GOE matrix, while $Q_-$ belongs to class D, as considered in the main text. As before, we can express each moment of $Q_\pm$ in terms of the propagator of $Q$ using Wick contraction. The second moment is given by

$$\left\langle \mathrm{Tr}\, Q_\pm^2 \right\rangle = \left\langle \mathrm{Tr}\, Q^2 \right\rangle \pm \left\langle \mathrm{Tr}\, QQ^T \right\rangle = \left\langle \mathrm{Tr}\, Q^2 \right\rangle \pm \sum_{ab} \langle Q_{ab} Q_{ab} \rangle = N_F \pm 1 . \tag{C.5}$$

We see that the contribution due to (anti)symmetrization is subleading in $1/N_F$. This carries over to higher moments: any contraction of $Q$ and $Q^T$ is suppressed by $1/N_F$. We note that, incidentally, the corrections due to (anti)symmetrization are less suppressed than those arising due to nonplanarity. We can proceed similarly for the fourth moment, which is given by:

$$\left\langle \mathrm{Tr}\, Q_\pm^4 \right\rangle = \frac{1}{2} \left\langle \mathrm{Tr}\, Q^4 \pm 4\,\mathrm{Tr}\, Q^3 Q^T + 2\,\mathrm{Tr}\, Q^2 (Q^T)^2 + \mathrm{Tr}\, QQ^T QQ^T \right\rangle . \tag{C.6}$$

The only unsuppressed contractions (either by symmetrization or nonplanarity) are

$$\frac{1}{2} \left( 2\langle \mathrm{Tr}\, \overset{\frown}{Q}\overset{\frown}{Q}\overset{\frown}{Q}\overset{\frown}{Q} \rangle + 2\langle \mathrm{Tr}\, \overset{\frown}{Q}\overset{\frown}{Q}\overset{\frown}{Q^T}\overset{\frown}{Q^T} \rangle \right) = \left\langle \mathrm{Tr}\, Q^4 \right\rangle . \tag{C.7}$$

For the sixth moment, we find

$$\frac{1}{4} \left( 5\langle \mathrm{Tr}\, \overset{\frown}{Q}\overset{\frown}{Q}\overset{\frown}{Q}\overset{\frown}{Q}\overset{\frown}{Q}\overset{\frown}{Q} \rangle + 15\langle \mathrm{Tr}\, \overset{\frown}{Q}\overset{\frown}{Q}\overset{\frown}{Q}\overset{\frown}{Q}\overset{\frown}{Q^T}\overset{\frown}{Q^T} \rangle \right) = \left\langle \mathrm{Tr}\, Q^6 \right\rangle , \tag{C.8}$$

and similarly for higher moments. We conclude that $\left\langle \mathrm{Tr}\, Q_\pm^p \right\rangle = \left\langle \mathrm{Tr}\, Q^p \right\rangle$ and, hence, their resolvents coincide.

## D  Spectral gap from the asymptotic analysis of Eq. (36)

In this appendix, we extract the leading-order behavior of the gap from an asymptotic analysis of Eq. (36), which we reproduce here for convenience,

$$y^2 = -\frac{m}{4gx} - \left( \frac{g/2}{1 - 4gx} + \frac{m}{4x} - \frac{1}{4g} \right)^2 ,$$

in the limits $g \to \infty$ and $g \to 0$. This procedure can be employed as an alternative to the perturbation theory of Sec. 3.2 and yields not only the leading scaling with $g$ but also the exact prefactor.

Let us fix the notation first. We say that $A(g) \asymp B(g)$ if the leading order terms of $A$ and $B$ are equal (more rigorously, $\lim_{g \to \infty} A(g)/B(g) = 1$). We are therefore interested in computing

$C$ and $\alpha$ in $x_g(g) \asymp -Cg^{\alpha}$, where $x_g$ is (minus) the gap and $C \geq 0$. We can now proceed as described before, setting $y = 0$ and replacing $x$ by $-Cg^{\alpha}$ in order to estimate the asymptotic behavior in Eq. (36). After some manipulations, it becomes:

$$\pm\left(\sqrt{\frac{m}{4C}}g^{-(\alpha+1)/2} + \sqrt{4mC}\,g^{(\alpha+1)/2}\right) \asymp \left(\frac{1}{2}-m\right)g - Cg^{\alpha} - \frac{m}{4C}g^{-\alpha} - \frac{1}{4}g^{-1}. \qquad (D.1)$$

If none of the exponents of $g$ in the previous equation match, then every term must vanish identically and the equation becomes trivial. There are five different values of $\alpha$ for which some of the exponents match: $\alpha = 1$, $\alpha = 0$, $\alpha = -1/3$, $\alpha = -1$, and $\alpha = -3$.

- At $\alpha = -3$ the leading order term on the right-hand side is $-m/(4C)g^3$ and it is unmatched on the left-hand side, which implies that $m = 0$. Since the gap at $m = 0$ trivially vanishes, we must rule out $\alpha = -3$.

- Similarly, for $\alpha = 0$, we obtain that $m = 1/2$ and then $C = 0$, which also means that $\alpha = 0$ is not allowed.

- For $\alpha = -1$, Eq. (D.1) becomes

$$\frac{1}{2} - m - \frac{m}{4C} = 0 \Leftrightarrow C = \frac{m/4}{1/2-m}. \qquad (D.2)$$

  Note that this equation is only valid for $m < 1/2$ as $C$ must be positive.

- If we instead assume $\alpha = -1/3$, then $m = 1/2$ and $\pm\sqrt{4mC} = -m/(4C)$ $\Leftrightarrow C^{3/2} = \pm\sqrt{m}/8$. Since $C$ is positive, $C = m^{1/3}/4 = 2^{-1/3}/4$.

- Last but not least, for $\alpha = 1$, $\pm\sqrt{4mC} = 1/2 - m - C \Leftrightarrow C = \left(1 \pm \sqrt{2m}\right)^2/2$. These solutions exist for all values of $m$.

By inspection of the graphics present in Fig. 2 of the main text, we can assign one of the asymptotic behaviors above to each of the intersection points of the boundary with $y = 0$.

- For $m < 1/2$, the system is in phase II and thus there are two disconnected regions in the spectrum and four intersection points. The ones that delimit the region of eigenvalues with smaller real part correspond to the two solutions for the case $\alpha = 1$. The other two must correspond to the other solution available, at $\alpha = -1$, which means that they have the same scaling behavior with $g$. Since the gap is given by (minus) the intersection point with the largest real part, we conclude that for $m < 1/2$, Gap $\sim \frac{m/4}{1/2-m}g^{-1}$.

- For $m \geq 1/2$ there is just a single connected region (phase I) and thus there are just two intersection points. At $m = 1/2$, Gap $\sim 2^{-1/3}g^{-1/3}/4$.

- For $m > 1/2$, Gap $\sim \left(1 - \sqrt{2m}\right)^2 g/2$, in agreement with the Marchecko-Pastur formula, Eq. (40).

The behaviour of $x_g(g)$ in the limit $g \to 0$ can also be extracted from Eq. (D.1) (note that now $A(g) \asymp B(g)$ means that $\lim_{g \to 0} A(g)/B(g) = 1$). The non-positive solutions of $\alpha$ cannot represent the asymptotic behavior of the gap, since it is clear from the expression for $K(g)$ that the gap converges to 0 in the limit $g \to 0$. Therefore, we are left with $\alpha = 1$, for which the condition $\pm\sqrt{m/(4C)} = -1/4 - m/(4C)$ must be verified. Since $C > 0$, this equation becomes $\sqrt{m/(4C)} = 1/2 \Leftrightarrow C = m$ and thus Gap $\sim mg$, in agreement with perturbation theory.

# E  Perturbative steady-state spectrum

In this appendix, we compute perturbatively the steady-state correlation matrix and its spectrum, in the limits of very weak and very strong dissipation. From Eqs. (B.17) and (B.32) in Appendix B, we can write the following equation for the correlation matrix of the steady state:

$$\chi K^\dagger - K\chi = igN\,. \tag{E.1}$$

We are interested in determining the solution to Eq. (E.1) in the limits $g \to 0$ and $g \to \infty$.

## E.1  Weak dissipation

In the weak dissipation limit, we expand $\chi$ in powers of $g$, $\chi = \chi^{(0)} + \chi^{(1)}g + \chi^{(2)}g^2 + \dots$, and plug it in Eq. (E.1). Comparing terms order by order, we obtain that:

$$\begin{aligned}
\left[\chi^{(0)}, H\right] &= 0\,, \\
-i\left[\chi^{(1)}, H\right] + \Gamma\chi^{(0)} + \chi^{(0)}\Gamma &= N\,.
\end{aligned} \tag{E.2}$$

Let $\{v_i\}_i$ be an orthonormal eigenbasis of $H$ (the element $v_j$ is to be understood as a column vector). Since $\chi^{(0)}$ commutes with $H$, $\{v_i\}_i$ can always be chosen to also form an eigenbasis of $\chi^{(0)}$. Thus, denoting the eigenvalues of $H$ and $\chi^{(0)}$ as $\epsilon_i$ and $\lambda_i$, respectively, the second equality in Eq. (E.2) yields

$$-i\left(\chi_{i,i}^{(1)}\epsilon_i - \chi_{i,i}^{(1)}\epsilon_i\right) + 2\lambda_i v_i^\dagger \Gamma v_i = v_i^\dagger N v_i \iff \lambda_i = \frac{1}{2}\frac{v_i^\dagger N v_i}{v_i^\dagger \Gamma v_i}\,, \tag{E.3}$$

where $\chi_{i,i}^{(1)}$ is short-hand notation for $v_i^\dagger \chi^{(1)} v_i$. Note that we used that both $H$ and $\chi^{(0)}$ are Hermitian and, consequently, their spectrum is real. We thus find that

$$\lim_{g \to 0} \chi(g) = \chi^{(0)} = \sum_i \lambda_i v_i v_i^\dagger\,, \tag{E.4}$$

with $\lambda_i$ given by Eq. (E.3). Note that, in general, we could have to be more careful here. If some of the $v_i^\dagger \Gamma v_i$ were zero, this result would not be valid and we would have to look at terms of higher order in $g$ in the power expansion of Eq. (E.1). This could only happen if $v_i$ belongs to the nullspace of $\Gamma$, since $\Gamma$ is a positive semidefinite matrix. However, since in the present paper we work with random and independent $H$ and $\Gamma$, the probability of $v_i^\dagger \Gamma v_i = 0$ is zero as the nullspace of $\Gamma$ is a set of measure zero.

## E.2  Strong dissipation

In the strong dissipation limit, $\chi$ is expanded in powers of $1/g$, $\chi = \chi^{(0)} + \chi^{(1)}/g + \chi^{(2)}/g^2 + \dots$ and the comparison of terms of the same order in Eq. (E.1) yields

$$\begin{aligned}
\Gamma\chi^{(0)} + \chi^{(0)}\Gamma &= N\,, \\
\Gamma\chi^{(1)} + \chi^{(1)}\Gamma &= i\left[\chi^{(0)}, H\right]\,, \\
\Gamma\chi^{(2)} + \chi^{(2)}\Gamma &= i\left[\chi^{(1)}, H\right]\,.
\end{aligned} \tag{E.5}$$

Let $\{\gamma_i\}_i$ and $\{w_i\}_i$ be the set of eigenvalues and orthonormal eigenvectors of $\Gamma$, respectively. Then, it is easy to see that from the first equality in Eq. (E.5), we can obtain

$$\left(\gamma_i + \gamma_j\right)\chi_{i,j}^{(0)} = w_i^\dagger N w_j \iff \chi_{i,j}^{(0)} = \frac{w_i^\dagger N w_j}{\gamma_i + \gamma_j}\,, \quad \text{if } \gamma_i + \gamma_j \neq 0\,, \tag{E.6}$$

where $\chi_{i,j}^{(0)} = w_i^\dagger \chi^{(0)} w_j$. Since $\Gamma$ is positive semidefinite, $\gamma_i \geq 0$. If all $\gamma_i$ are positive, then Eq. (E.6) completely determines $\chi^{(0)} = \lim_{g \to \infty} \chi(g)$.

However, if $\gamma_i + \gamma_j = 0$, Eq. (E.6) trivially holds for all $\chi_{i,j}^{(0)}$ and we have to look at higher order terms to compute it. Note that the nullspace of the matrix $\Gamma$ is contained in the nullspace of $N$. In fact, since $N$ and $N_S = SN^T S$ are positive semidefinite matrices, $w_i^\dagger N w_i \geq 0$ and $w_i^\dagger N_S w_i \geq 0$. If $w_i$ belongs to the nullspace of $\Gamma$, then $w_i^\dagger \Gamma w_i = 0 \Rightarrow w_i^\dagger N w_i = 0 \Rightarrow N w_i = 0$.

Suppose, for example, that, in Eq. (E.6), $\gamma_j = 0$. Then, because of the property we have just shown, $\chi_{i,j}^{(0)} = 0$. This means that, introducing the projector onto the nullspace of $\Gamma$, $P_N$, and onto its orthogonal complement, $P_{\bar{N}}$, we have $\chi_{N\bar{N}}^{(0)} = P_N \chi^{(0)} P_{\bar{N}} = 0$ and $\chi_{\bar{N}N}^{(0)} = P_{\bar{N}} \chi^{(0)} P_N = 0$. This is equivalent to the statement that $\chi^{(0)}$ is block diagonal,

$$\chi^{(0)} = \begin{bmatrix} \chi_{\bar{N}\bar{N}}^{(0)} & 0 \\ 0 & \chi_{NN}^{(0)} \end{bmatrix}, \tag{E.7}$$

where $\chi_{\bar{N}\bar{N}}^{(0)} = P_{\bar{N}} \chi^{(0)} P_{\bar{N}}$ and $\chi_{NN}^{(0)} = P_N \chi^{(0)} P_N$. $\chi_{\bar{N}\bar{N}}^{(0)}$ can be determined directly from Eq. (E.6),

$$\left( \chi_{\bar{N}\bar{N}}^{(0)} \right)_{ij} = \frac{\left( w_{\bar{N}} \right)_i^\dagger N_{\bar{N}\bar{N}} \left( w_{\bar{N}} \right)_j}{\gamma_i + \gamma_j}, \tag{E.8}$$

with $(w_{\bar{N}})_i$ an eigenvector of $\Gamma$ that does not belong to its nullspace, whereas, to compute $\chi_{NN}^{(0)}$, we must resort to the other equalities in Eq. (E.5). From the second one, we can write

$$\begin{aligned} &\chi_{NN}^{(0)} H_{NN} - H_{NN} \chi_{NN}^{(0)} = 0, \\ &\chi_{\bar{N}\bar{N}}^{(0)} H_{\bar{N}N} - H_{\bar{N}N} \chi_{NN}^{(0)} + i\Gamma_{\bar{N}\bar{N}} \chi_{\bar{N}N}^{(1)} = 0, \\ &\chi_{\bar{N}\bar{N}}^{(0)} H_{\bar{N}\bar{N}} - H_{\bar{N}\bar{N}} \chi_{\bar{N}\bar{N}}^{(0)} + i\Gamma_{\bar{N}\bar{N}} \chi_{\bar{N}\bar{N}}^{(1)} + i\chi_{\bar{N}\bar{N}}^{(1)} \Gamma_{\bar{N}\bar{N}} = 0, \end{aligned} \tag{E.9}$$

where we used that $\Gamma_{\bar{N}N} = \Gamma_{NN} = \chi_{N\bar{N}}^{(0)} = 0$. The first equality states that $\chi_{NN}^{(0)}$ and $H_{NN}$ share the same eigenbasis but it does not allow us to compute its eigenvalues. To determine them, we start by observing that, from the third equality in Eq. (E.5),

$$\chi_{NN}^{(1)} H_{NN} - H_{NN} \chi_{NN}^{(1)} = H_{N\bar{N}} \chi_{\bar{N}N}^{(1)} - \chi_{N\bar{N}}^{(1)} H_{\bar{N}N}. \tag{E.10}$$

Up to this point, we were completely free to choose a basis $\{(w_N)_i\}_i$ for the nullspace of $\Gamma$ as long as we kept it orthonormal. However, we will fix it now in such a way that $H_{NN}$ becomes a diagonal matrix: $H_{NN} = \sum_i \epsilon_i (w_N)_i (w_N)_i^\dagger$ (note that it is always possible since $H_{NN}$ is Hermitian). Plugging this in Eq. (E.10), we arrive at:

$$\left( \chi_{NN}^{(1)} \right)_{j,k} (\epsilon_k - \epsilon_j) = \left( H_{N\bar{N}} \chi_{\bar{N}N}^{(1)} - \chi_{N\bar{N}}^{(1)} H_{\bar{N}N} \right)_{jk}. \tag{E.11}$$

Setting $j = k$, we can eliminate the new variable $\chi_{NN}^{(1)}$ and write a closed equation for the variables already present in Eq. (E.9):

$$\left( H_{N\bar{N}} \chi_{\bar{N}N}^{(1)} - \chi_{N\bar{N}}^{(1)} H_{\bar{N}N} \right)_{jj} = 0. \tag{E.12}$$

We can now solve the second equality in Eq. (E.9) for $\chi_{N\bar{N}}^{(1)}$ (and its Hermitian conjugate), insert it in the last equality, and use the fact that, since $\chi_{NN}^{(0)} H_{NN} = H_{NN} \chi_{NN}^{(0)}$, $\{(w_N)_i\}_i$ is also an eigenbasis of $\chi_{NN}^{(0)}$, i.e., $\chi_{NN}^{(0)} (w_N)_i = \lambda_i (w_N)_i$:

$$\left( -i H_{N\bar{N}} \Gamma_{\bar{N}\bar{N}}^{-1} H_{\bar{N}N} \chi_{NN}^{(0)} + i H_{N\bar{N}} \Gamma_{\bar{N}\bar{N}}^{-1} \chi_{\bar{N}\bar{N}}^{(0)} H_{\bar{N}N} - i \chi_{NN}^{(0)} H_{N\bar{N}} \Gamma_{\bar{N}\bar{N}}^{-1} H_{\bar{N}N} + i H_{N\bar{N}} \chi_{\bar{N}\bar{N}}^{(0)} \Gamma_{\bar{N}\bar{N}}^{-1} H_{\bar{N}N} \right)_{jj} = 0, \tag{E.13}$$

which implies:

$$2\sum_{\beta}\left(H_{N\bar{N}}\right)_{j\beta}\gamma_{\beta}^{-1}\left(H_{\bar{N}N}\right)_{\beta j}\lambda_{j}=\sum_{\alpha\beta}\left(\gamma_{\alpha}^{-1}+\gamma_{\beta}^{-1}\right)\left(H_{N\bar{N}}\right)_{j\beta}\left(\chi_{\bar{N}\bar{N}}^{(0)}\right)_{\beta\alpha}\left(H_{\bar{N}N}\right)_{\alpha j}. \tag{E.14}$$

In the preceding equations, the Greek indices label the eigenvectors that do not belong to the nullspace of $\Gamma$ and $\left(H_{N\bar{N}}\right)_{j\beta}=(w_{N})_{j}^{\dagger}H\left(w_{\bar{N}}\right)_{\beta}$. We can now use Eq. (E.8) to simplify the above result,

$$2\sum_{\beta}\left(H_{N\bar{N}}\right)_{j\beta}\gamma_{\beta}^{-1}\left(H_{\bar{N}N}\right)_{\beta j}\lambda_{j}=\sum_{\alpha\beta}\left(H_{N\bar{N}}\right)_{j\beta}\frac{\left(N_{\bar{N}\bar{N}}\right)_{\beta\alpha}}{\gamma_{\alpha}\gamma_{\beta}}\left(H_{\bar{N}N}\right)_{\alpha j}, \tag{E.15}$$

and, solving for $\lambda_{j}$:

$$\lambda_{j}=\frac{1}{2}\frac{(w_{N})_{j}^{\dagger}H_{N\bar{N}}\Gamma_{\bar{N}\bar{N}}^{-1}N_{\bar{N}\bar{N}}\Gamma_{\bar{N}\bar{N}}^{-1}H_{\bar{N}N}(w_{N})_{j}}{(w_{N})_{j}^{\dagger}H_{N\bar{N}}\Gamma_{\bar{N}\bar{N}}^{-1}H_{\bar{N}N}(w_{N})_{j}}. \tag{E.16}$$

In conclusion, we found that

$$\lim_{g\to\infty}\chi(g)=\sum_{\alpha\beta}\left(w_{\bar{N}}\right)_{\alpha}\frac{\left(w_{\bar{N}}\right)_{\alpha}^{\dagger}N_{\bar{N}\bar{N}}\left(w_{\bar{N}}\right)_{\beta}}{\gamma_{\alpha}+\gamma_{\beta}}\left(w_{\bar{N}}\right)_{\beta}^{\dagger}+\sum_{i}\lambda_{i}(w_{N})_{i}(w_{N})_{i}^{\dagger}. \tag{E.17}$$

Considering the case $m<1/2$, we now show that Eq. (E.8) can be considerably simplified. Noting that $N=\Gamma+\Gamma_{B}$, it is easy to see that

$$\left(\chi_{\bar{N}\bar{N}}^{(0)}\right)_{ij}=\frac{1}{2}\delta_{i,j}+\frac{\left(w_{\bar{N}}\right)_{i}^{\dagger}\Gamma_{B}\left(w_{\bar{N}}\right)_{j}}{\gamma_{i}+\gamma_{j}}. \tag{E.18}$$

If we now decompose $l_{\mu,j}=l_{\mu,j}^{R}+il_{\mu,j}^{I}$ into its real and imaginary parts, $\Gamma$ and $\Gamma_{B}$ can be rewritten as

$$\Gamma_{j,k}=\sum_{\mu}^{M}\left(l_{\mu,j}^{R}l_{\mu,k}^{R}+l_{\mu,j}^{I}l_{\mu,k}^{I}\right)=\sum_{\mu}^{2M}y_{\mu,j}y_{\mu,k}, \tag{E.19}$$
$$(\Gamma_{B})_{j,k}=i\sum_{\mu}^{M}\left(l_{\mu,j}^{R}l_{\mu,k}^{I}-l_{\mu,j}^{I}l_{\mu,k}^{R}\right)=\sum_{\mu\nu}^{2M}y_{\mu,j}\mathcal{J}_{\mu,\nu}y_{\nu,k},$$

where $\mathcal{J}=\begin{bmatrix}0 & i\mathbb{1}\\-i\mathbb{1} & 0\end{bmatrix}$ and $y=\begin{bmatrix}l^{R}\\l^{I}\end{bmatrix}$.

Since we assume that $2M<2N_{F}$, and due to Eq. (E.18), only the subspace spanned by the eigenvectors of $\Gamma$ associated with nonzero eigenvalues is relevant. We, therefore, restrict all the following analysis to this subspace. We start by changing coordinates to the basis defined by $y_{\alpha}$, writing the eigenvectors of $\Gamma$ as $\left(w_{\bar{N}}\right)_{i}=\sum_{\alpha}\left(\tilde{w}_{\bar{N}}\right)_{i,\alpha}y_{\alpha}$.

The action of $\Gamma$ on a vector $x_{j}=\sum_{\alpha}\tilde{x}_{\alpha}y_{\alpha,j}$ that belongs to this subspace can thus be written as

$$\sum_{j}\Gamma_{i,j}x_{j}=\sum_{\mu}^{2M}y_{\mu,j}\sum_{\alpha}\sum_{k}y_{\mu,k}y_{\alpha,k}\tilde{x}_{\alpha}=\sum_{\mu}^{2M}y_{\mu,j}\sum_{\alpha}\tilde{\Gamma}_{\mu,\alpha}\tilde{x}_{\alpha}, \tag{E.20}$$

implying that $\tilde{\Gamma}_{\mu,\alpha}=\sum_{k}y_{\mu,k}y_{\alpha,k}$ is the matrix representation of $\Gamma$ in the new basis.[8] Similarly, it is easy to see that $\Gamma_{B}$ in the new basis reads $\tilde{\Gamma}_{B}=\mathcal{J}\tilde{\Gamma}$.

---

[8]Note that in the case where the entries $y_{\mu,k}$ are independent real random variables, this transformation establishes an equivalence between the non-zero eigenvalues of a Wishart matrix with dimension $2N_{F}>2M$ and $2M$ degrees of freedom with the spectrum of another Wishart matrix with dimension $2M$ and $2N_{F}$ degrees of freedom.

Since the matrix that implements this change of basis is not orthonormal, the metric after the transformation is no longer the identity. In fact, it becomes $\tilde{\Gamma}$, which means that it must be included in all inner products computed in this basis. This allows us to rewrite Eq. (E.18) as

$$\left(\chi_{\tilde{N}\tilde{N}}^{(0)}\right)_{ij} = \frac{1}{2}\delta_{i,j} + \frac{\left(\tilde{w}_{\tilde{N}}\right)_i^T \tilde{\Gamma}\mathcal{J}\tilde{\Gamma}\left(\tilde{w}_{\tilde{N}}\right)_j}{\gamma_i + \gamma_j} = \frac{1}{2}\delta_{i,j} + \gamma_i\gamma_j\frac{\left(\tilde{w}_{\tilde{N}}\right)_i^T \mathcal{J}\left(\tilde{w}_{\tilde{N}}\right)_j}{\gamma_i + \gamma_j}. \tag{E.21}$$

Note that, since the eigenvectors $\left(w_{\tilde{N}}\right)_i$ are normalized, we have

$$\left(\tilde{w}_{\tilde{N}}\right)_i^T \tilde{\Gamma}\left(\tilde{w}_{\tilde{N}}\right)_i = 1 \Longleftrightarrow \left(\tilde{w}_{\tilde{N}}\right)_i^T \left(\tilde{w}_{\tilde{N}}\right)_i = \frac{1}{\gamma_i}, \tag{E.22}$$

which means that the norm of $\left(\tilde{w}_{\tilde{N}}\right)_i$ under the usual inner product is $1/\sqrt{\gamma_i}$. For convenience, we perform the transformation $\left(\tilde{w}_{\tilde{N}}\right)_i \rightarrow \left(\tilde{w}_{\tilde{N}}\right)_i /\sqrt{\gamma_i}$, leading to our final expression for $\left(\chi_{\tilde{N}\tilde{N}}^{(0)}\right)_{ij}$:

$$\left(\chi_{\tilde{N}\tilde{N}}^{(0)}\right)_{ij} = \frac{1}{2}\delta_{i,j} + \sqrt{\gamma_i\gamma_j}\frac{\left(\tilde{w}_{\tilde{N}}\right)_i^T \mathcal{J}\left(\tilde{w}_{\tilde{N}}\right)_j}{\gamma_i + \gamma_j}. \tag{E.23}$$

# F  GOE statistics for the steady state in the limit $g \rightarrow \infty$, $m \rightarrow 0$

In this subsection, we are interested in studying the limit $m \rightarrow 0$ of Eq. (E.23) for the case of a random Liouvillian (sampled as described in Sec. 2). We show that the steady state supports GOE statistics in this limit.

Assuming $M$ and $N_F$ are sufficiently large, but such that $M \ll 2N_F$, we can write

$$\tilde{\Gamma} = \mathbb{1} + \sqrt{\frac{M}{2N_F}}\tilde{\Gamma}^{(1)}, \tag{F.1}$$

where, from the central limit theorem, the spectrum of $\tilde{\Gamma}^{(1)}$, $\gamma_i^{(1)}$ (with $\gamma_i = 1 + \sqrt{m}\gamma_i^{(1)}$), is convergent in the limit $M \rightarrow \infty$ and $N_F \rightarrow \infty$ and of order one. Plugging this in Eq. (E.23) and Taylor expanding it up to second order in $\sqrt{m} = \sqrt{M/(2N_F)}$, we obtain:

$$\left(\chi_{\tilde{N}\tilde{N}}^{(0)}\right)_{ij} = \frac{1}{2}\delta_{i,j} + \frac{1}{2}\tilde{w}_i^T \mathcal{J}\tilde{w}_j - \frac{m}{16}\left(\gamma_i^{(1)} - \gamma_j^{(1)}\right)^2 \tilde{w}_i^T \mathcal{J}\tilde{w}_j + \mathcal{O}\left(m^{3/2}\right) = \tilde{w}^T \chi_{\tilde{N}\tilde{N}}^{(w)}\tilde{w} + \mathcal{O}\left(m^{3/2}\right), \tag{F.2}$$

where we defined

$$\chi_{\tilde{N}\tilde{N}}^{(w)} = \frac{1}{2}\mathbb{1} + \frac{1}{2}\mathcal{J} - \frac{m}{16}\left(\left(\tilde{\Gamma}^{(1)}\right)^2 \mathcal{J} + \mathcal{J}\left(\tilde{\Gamma}^{(1)}\right)^2 - 2\left(\tilde{\Gamma}^{(1)}\right)\mathcal{J}\left(\tilde{\Gamma}^{(1)}\right)\right). \tag{F.3}$$

We conclude that the diagonalization of $\chi_{\tilde{N}\tilde{N}}^{(w)}$ provides the eigenvalues of $\chi_{\tilde{N}\tilde{N}}^{(0)}$ up to first order in $m$.

We now resort to first-order perturbation theory to simplify the previous equation while keeping it exact to first order in $m$. The eigenvalues of $\mathcal{J}$ are either $-1$ or $1$, which means that the spectrum of $\chi_{\tilde{N}\tilde{N}}^{(w)}$ splits into two regions, one close to 0 and the other to 1. $\mathcal{J}$ is diagonalized by the matrix

$$\tilde{U} = \frac{1}{\sqrt{2}}\begin{bmatrix} \mathbb{1} & \mathbb{1} \\ -i\mathbb{1} & i\mathbb{1} \end{bmatrix}. \tag{F.4}$$

Defining

$$
\tilde{\Gamma}_{\tilde{U}}^{(1)} = \tilde{U}\tilde{\Gamma}^{(1)}\tilde{U}^{\dagger} = \begin{bmatrix} \left(\tilde{\Gamma}_{\tilde{U}}^{(1)}\right)_{11} & \left(\tilde{\Gamma}_{\tilde{U}}^{(1)}\right)_{12} \\ \left(\tilde{\Gamma}_{\tilde{U}}^{(1)}\right)_{21} & \left(\tilde{\Gamma}_{\tilde{U}}^{(1)}\right)_{22} \end{bmatrix}
$$

$$
= \frac{1}{2}\begin{bmatrix} \tilde{\Gamma}_{11}^{(1)} + \tilde{\Gamma}_{22}^{(1)} + i\left(\tilde{\Gamma}_{21}^{(1)} - \tilde{\Gamma}_{12}^{(1)}\right) & \tilde{\Gamma}_{11}^{(1)} - \tilde{\Gamma}_{22}^{(1)} + i\left(\tilde{\Gamma}_{21}^{(1)} + \tilde{\Gamma}_{12}^{(1)}\right) \\ \tilde{\Gamma}_{11}^{(1)} - \tilde{\Gamma}_{22}^{(1)} - i\left(\tilde{\Gamma}_{21}^{(1)} + \tilde{\Gamma}_{12}^{(1)}\right) & \tilde{\Gamma}_{11}^{(1)} + \tilde{\Gamma}_{22}^{(1)} - i\left(\tilde{\Gamma}_{21}^{(1)} - \tilde{\Gamma}_{12}^{(1)}\right) \end{bmatrix}, \tag{F.5}
$$

the first order correction to the positive eigenvalues of $\mathcal{J}$ is given by the spectrum of

$$
-\frac{m}{8}\left(\tilde{\Gamma}_{\tilde{U}}^{(1)}\right)_{12}\left(\tilde{\Gamma}_{\tilde{U}}^{(1)}\right)_{21} = -\frac{m}{16}\left(\tilde{\Gamma}_{11}^{(1)} - \tilde{\Gamma}_{22}^{(1)} + i\left(\tilde{\Gamma}_{21}^{(1)} + \tilde{\Gamma}_{12}^{(1)}\right)\right)\left(\tilde{\Gamma}_{11}^{(1)} - \tilde{\Gamma}_{22}^{(1)} - i\left(\tilde{\Gamma}_{21}^{(1)} + \tilde{\Gamma}_{12}^{(1)}\right)\right)
$$

$$
= -\frac{m}{16}VV^{\dagger}, \tag{F.6}
$$

where $V = \tilde{\Gamma}_{11}^{(1)} - \tilde{\Gamma}_{22}^{(1)} + i\left(\tilde{\Gamma}_{21}^{(1)} + \tilde{\Gamma}_{12}^{(1)}\right)$ is a generic complex symmetric matrix. In the limit of $m \to 0$ the spacings of the eigenvalues of $\chi_{\tilde{N}\tilde{N}}^{(0)}$ are thus determined by the spacing of the eigenvalues of $VV^{\dagger}$. Since the eigenvalues of $VV^{\dagger}$ coincide with the positive eigenvalues of the chiral matrix

$$
\begin{bmatrix} 0 & V^{\dagger} \\ V & 0 \end{bmatrix}, \tag{F.7}
$$

$VV^{\dagger}$ belongs to class CI [56, 64], which has the same level statistics as the GOE. This finally explains the observation that the level correlations of Eq. (E.23) approach those of the GOE in the limit $m \to 0$.

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
