# Peer review of "Spectral and steady-state properties of fermionic random quadratic Liouvillians"

_SciPost Physics, doi:SciPost Phys. 15, 145 (2023)_

## Round 1 · Referee Report · Anonymous · 2023-4-19

Strengths
Very beautiful and insightful results, which are (mostly) clearly explained.
Report
I have read with much interest this work by Costa et al. on the "Spectral and steady-state properties of fermionic random quadratic Liouvillians".
The authors found some nice results, notably that depending on number of jump operators (relative to the size of the system), one can observe two different phases. In one the rapidities can be grouped in a single region, while in the other case there are two distinct regions, each with very different properties. This occurs in this particular class of Hamiltonians they have considered.
I recommend publication of this article.
Here are some comments to be considered:
- BGS conjecture. Before this paper, another one was published, G. Casati, F. Valz-Gris, and I. Guarnieri, Connection between quantization of nonintegrable systems and statistical theory of spectra, Lett. Nuovo Cimento 28, 279 (1980), which should be mentioned
- the authors mention the paper from Sudarshan, Gorini, Kossakowski, but they refer to the master equation simply as Lindblad instead of GKSL as more commmon now, and maybe fair, in the literature
- In Eq.(19) you write max because the imaginary part of beta is negative?
- labels and axis in Fig.1 be better
- "The rate at which the system relaxes to the steady state is dictated by the Liouvillian spectral gap" In large systems it is possible that many exponential decays "add up" to give algebraic relaxation, although, of course, for finite systems it will eventually get back to an exponential. The authors can consider say something about this
- typo: "it is convenient to recast the it as a matrix acting"
- typo in caption of Fig.2 "The the 2N_F blue"
- typo after Eq.(39). "and Eq. 39)"
- Fig.3: the choice of colors for the lines could be improved, and the dashed lines are
- Figs.4 and 5 make me wonder: do you really need such extreme values of g to see such nice convergence to analytical predictions?
- The caption of Fig.9 is not very clear. Certainly an "(a)" is missing, but also it could be written more clearly so that it can be read more independently
Author: João Costa on 2023-06-24 [id 3760]
(in reply to Report 1 on 2023-04-19)
We thank Referee 1 for carefully reading our paper and for their comments, which we now address.
(We reproduce the referee's comments in quotation marks and give our answer below each comment.)
-"I have read with much interest this work by Costa et al. on the "Spectral and steady-state properties of fermionic random quadratic Liouvillians". The authors found some nice results, notably that depending on number of jump operators (relative to the size of the system), one can observe two different phases. In one the rapidities can be grouped in a single region, while in the other case there are two distinct regions, each with very different properties. This occurs in this particular class of Hamiltonians they have considered. I recommend publication of this article. ":
We thank the referee for their summary of our work and their very positive assessment.
-"Here are some comments to be considered:
- BGS conjecture. Before this paper, another one was published, G. Casati, F. Valz-Gris, and I. Guarnieri, Connection between quantization of nonintegrable systems and statistical theory of spectra, Lett. Nuovo Cimento 28, 279 (1980), which should be mentioned ":
We thank the referee for mentioning this paper. We have now referenced it in our article.
-"2. The authors mention the paper from Sudarshan, Gorini, Kossakowski, but they refer to the master equation simply as Lindblad instead of GKSL as more commmon now, and maybe fair, in the literature.":
We thank the referee for alerting us to this nomenclature issue. We have included the complete name of the equation the first time we mention it, although we keep the name Lindblad equation throughout the rest of the article, following standard practice.
-"3. In Eq.(19) you write max because the imaginary part of beta is negative? ":
Exactly. Since we agree that it is probably not the most intuitive way to write the expression for the gap, we have changed it to the minimum of the absolute value. We have applied the same change to Eq.~(5).
-"4. labels and axis in Fig.1 be better.":
We thank the referee for the suggestion, but Fig.~1 is intended to be mainly schematic, where we give a very brief overview of our results. All the graphics that we included there are more thoroughly characterized and examined (with complete labels and axes) in other parts of the article (Figs.~1(b) and (c) in Fig.~2; Figs.~1(d) and (f) in Figs.~5(a) and (b); Figs.1~(e) and (g) in Fig.~9).
-"5. "The rate at which the system relaxes to the steady state is dictated by the Liouvillian spectral gap" In large systems it is possible that many exponential decays "add up" to give algebraic relaxation, although, of course, for finite systems it will eventually get back to an exponential. The authors can consider say something about this.":
We thank the referee for this insightful observation. We have included a footnote, where we warn the reader of this possibility, which can only happen in systems of infinite dimension.
More precisely, our footnote reads:
`` Though this is certainly true for finite systems, for systems of infinite dimension, the eigenmodes with eigenfrequencies close to the gap can add up and lead to an algebraic relaxation to the steady state. ''
-"6. typo: "it is convenient to recast the it as a matrix acting""
-"7. typo in caption of Fig.2 "The the $2N_F$ blue""
-"8. typo after Eq.(39). "and Eq. 39)" }":
We thank the referee for spotting these typos. We have corrected them.
-"9. Fig.3: the choice of colors for the lines could be improved, and the dashed lines are ":
We thank the referee for pointing out that Fig.~3 could be made clearer. We have changed the colors of the full and dashed lines.
-"10. Figs.4 and 5 make me wonder: do you really need such extreme values of g to see such nice convergence to analytical predictions? ":
A good intuition for the range of values of $g$ for which one can see a nice convergence to the analytical predictions is provided by Fig.~6: when the reduced purity stabilizes, the spectral distribution of the steady state also stabilizes, which means that $g$ must be chosen well inside the plateaus of the curves in this plot. Since there is no numerically advantageous choice of $g$, one can just pick any value inside these plateaus.
-"11. The caption of Fig.9 is not very clear. Certainly an "(a)" is missing, but also it could be written more clearly so that it can be read more independently":
We thank the referee for pointing out that Fig.~9 and its caption could also be made clearer. We have made a few changes to the way we display the information contained in this figure and we hope it can now be understood more clearly.
We again thank Referee 1 for carefully reading our paper. We hope that, with the clarifications provided above and the changes effected to the manuscript, they will now consider our work to be ready for publication in SciPost Physics.
Author: João Costa on 2023-06-24 [id 3759]
(in reply to Report 2 on 2023-05-07)We thank Referee 2 for carefully reading our paper and for their comments, which we now address.
(We reproduce the referee's comments in quotation marks and give our answer below each comment.)
-" The authors consider a quadratic fermionic system in the framework of Markovian master equations. Both the Hamiltonian part and the dissipative part of the Liouvillian are random. They find some interesting results as a function of the scaling variable m=$N_F/M$, with $N_F$ the number of fermions and M the number of Lindblad operators (dissipative channels). In particular, they show that a "phase transition" happens as a function of m. The two phases are distinguished by the behaviour of the spectrum of the Liouvillian. In one phase the spectrum has a single component, whereas in the other is made of two distinct components. This difference is reflected in the properties of the steady state.
This paper presents some interesting results and it is well organised. I definitely recommend publication in Scipost Physics. " :
We thank the referee for their summary of our work and their very positive assessment. We are pleased that the referee recommends publication in SciPost Physics.
-"I have one comment for the authors
They discuss the behaviour of the purity. However, since the purity is not a proper measure of entanglement it is not clear a priori whether the non monotonic behaviour that the authors observe is related to entanglement or not. They could comment on that. Perhaps in future work the could study the negativity or alternatively the mutual information, which although not a proper entanglement measure, in practice is sensitive to entanglement, as it has been clarified in. ":
We thank the referee for this very interesting comment. We agree that the purity is not a proper measure of entanglement and in order to study the latter in the case of mixed states one needs to resort to quantities such as the mutual information or the negativity. In particular, it would be interesting to check whether we still observe this non-monotonic behavior only for $m<0.5$. However, as the Referee points out, this falls out of the scope of the present article, which is mainly concerned with a characterization of the universal properties of the steady state, such as level correlations. Nevertheless, we agree that it is definitely a topic worth looking into in the future and thus we added a paragraph at the end of Sec.~4.2 discussing this idea.
We again thank Referee 2 for carefully reading our paper. We hope that, with the clarifications provided above and the changes effected to the manuscript, they will now consider our work to be ready for publication in SciPost Physics.

---

## Round 1 · Referee Report · Anonymous · 2023-5-7

Strengths
- Interesting results in a simplified yet paradigmatic model.
- Thorough investigation of the problem and clear organisation of the results.
Weaknesses
no major weaknesses
Report
The authors consider a quadratic fermionic system in the framework of Markovian master equations. Both the Hamiltonian part and the dissipative part of the Liouvillian are random. They find some interesting results as a function of the scaling variable m=N_F/M, with
N_F the number of fermions and M the number of Lindblad operators (dissipative channels). In particular, they show that a ``phase transition'' happens as a function of m. The two phases are distinguished by the behaviour of the spectrum of the Liouvillian. In one phase the spectrum has a single component, whereas in the other is made of two distinct components. This difference is reflected in the properties of the steady state.
This paper presents some interesting results and it is well organised. I definitely recommend publication in Scipost Physics.
I have one comment for the authors
They discuss the behaviour of the purity. However, since the purity is not a proper measure of entanglement it is not clear a priori whether the non monotonic behaviour that the authors observe is related to entanglement or not. They could comment on that. Perhaps in future work the could study the negativity or alternatively the mutual information, which although not a proper entanglement measure, in practice is sensitive to entanglement, as it has been clarified in
arXiv:2209.14164
Phys. Rev. B 105, 144305 (2022)

---

## Editorial Decision

published